# Robust Latent Neural Operators for a Family of Systems with Sparse Observations

## Abstract

Neural operator methods have achieved significant success in the efficient simulation and inverse problems of complex systems by learning a mapping between two infinite-dimensional Banach spaces. However, existing methods still exhibit room for optimization in terms of robustness and modeling accuracy. Specifically, existing methods are characterized by sensitivity to noise and a tendency to overlook the importance of sparse observations in new domains. Therefore, we propose a robust latent neural operator based on the variational autoencoder framework. In this method, an encoder based on recurrent neural networks effectively extracts sequential information and dynamical characteristics embedded in the domain-specific sparse observations. Subsequently, a neural operator in latent space and a decoder facilitate the modelling of the original system. Additionally, for certain higher-dimensional systems, opting for a lower-dimensional latent space can reduce task complexity while still maintaining satisfactory modeling performance. We conduct experiments across several representative systems, and the results validate that our method achieves superior modeling accuracy and enhanced robustness compared to the state of the art baseline approaches.

## 1 Introduction

Modeling and forecasting complex dynamical systems have emerged as pivotal research themes within the scientific machine learning domain, witnessing the development of numerous efficacious methods and tools (Tang et al., 2020; Wang et al., 2023a). These advancements play a crucial role across various domains of complex systems research, not only aiding in the understanding of the intricate behaviors and evolutionary patterns of real-world systems, but also laying the groundwork for downstream tasks such as system prediction (Li et al., 2024b;a), structure inference (Brugere et al., 2018; Lorch et al., 2022), change point detection (Bjørheim et al., 2022; Li et al., 2023), tipping point prediction (Bury et al., 2021; Grziwotz et al., 2023), and system control (Bertsekas, 2019; Wang et al., 2023b).

Traditional artificial models, such as prevalent ordinary differential equation (ODE) and partial differential equation (PDE) systems, facilitate the modeling of original systems through determinate rules. However, these methods often struggle to precisely model more intricate behaviors in real-world systems. This gap underlines an urgent need for data-driven and machine learning approaches to explore the dynamics. To identify critical terms within the dynamical equations, the Sparse Identification of Nonlinear Dynamical Systems (SINDy) (Brunton et al., 2016; Kaiser et al., 2018) approach is commonly employed. Subsequently, benefiting from the universal approximation capability of deep neural networks (Hornik, 1991), an increasing number of researchers are now focusing on neural network tools. For instance, Recurrent Neural Networks (RNNs) and their extended architectures (Memory, 2010; Cho et al., 2014; Suárez et al., 2024) have proven to be effective for processing sequential data; Neural Ordinary Differential Equations (NODEs) (Chen et al., 2018) can handle data with irregular intervals, thereby enabling continuous modeling; Graph Neural Networks (GNNs) (Murphy et al., 2021; Liu et al., 2023) are capable of efficiently managing data with graph structures. Although these approaches have achieved success in their specific domains, they necessitate retraining of the neural networks when environmental parameters change. Consequently, it is imperative to explore methods for modeling a family of systems.

In response to the above challenge, neural operator methods have been proposed in recent years (Lu et al., 2021; Li et al., 2020). These methods represent an emergent technology within the domain of deep learning, aimed at establishing a machine learning model capable of learning mappings from functional spaces to functional spaces. Specifically, neural operators can effectively handle conditions involving infinite-dimensional variations (including parametric functions, initial conditions, and boundary conditions, etc.) as inputs, and directly output future system states, thereby achieving modeling of a family of dynamical systems. Particularly for PDE systems, neural operators present significant advantages over traditional numerical resolution methods, such as finite element and finite difference methods. These advantages include enhanced solution speeds while maintaining accuracy, and more efficient handling of various inverse problems (Zhao et al., 2022; Wang & Wang, 2024).

However, existing neural operator approaches still have room for optimization in terms of noise robustness and modeling accuracy. For instance, traditional neural operator methods may underperform when the observational data is contaminated with noise. Moreover, in new testing environments, we might observe system states at several moments, which are often irregularly spaced. Current approaches overlook the crucial role of these sparse observational data from new domains, particularly the sequential information and dynamical characteristics embedded within them. Therefore, we introduce a more robust latent neural operator approach, termed as Robust Latent Neural Operator (RLNO), grounded in the variational autoencoder (VAE) framework. Specifically, the primary contributions of our work are as follows.

- RLNO utilizes more domain-specific information compared to traditional neural operators. When tested in new domains, our approach not only leverages sparse observational state information but also excavates and utilizes the dynamic information embedded within these sequential observations.

- RLNO represents a novel approach grounded in VAE framework, incorporating an RNN-based encoder. This method effectively harnesses the strengths of RNNs and neural operators to more adeptly extract and utilize the dynamical information within sparse observations from new domains.

- RLNO inherits characteristics from neural operator approaches, significantly surpassing the computational efficiency observed in RNN-based and Neural ODE-based methods. Furthermore, we design the OPERATOR-RNN encoder, which can encode unevenly spaced observations into the initial value distribution of the latent space more efficiently.

- RLNO can select a smaller latent space dimension, thereby reducing the complexity of operator learning, and facilitating the neural operator's ability to capture essential features with limited training data. Consequently, our approach can be extended to modeling tasks of higher-dimensional complex systems.

## 2 RELATED WORK

Recent years have seen broad interest and rapid advancements in the area of neural operators. Two of the most seminal contributions in this domain are the development of Deep Operator Network (DeepONet) introduced by Lu et al.(Lu et al., 2021), and Fourier Neural Operator (FNO) proposed by Li et al.(Li et al., 2020). These methods leverage the neural network model $\mathcal{G}_\theta$ to learn the operator mapping $\mathcal{G}^\dagger : \mathcal{U} \to \mathcal{S}$, where $\mathcal{U}$ and $\mathcal{S}$ are two infinite-dimensional Banach spaces. Here, our research focuses on the DeepONet framework, which primarily consists of two components: the trunk network and the branch network. Specifically, the trunk network accepts an arbitrary position $y$ as input and outputs the vector $\boldsymbol{c} = \{c_1, c_2, \cdots, c_m\}$. Concurrently, the branch network takes $l$ discrete observations $\{u(x_1), u(x_2), \cdots, u(x_l)\}$ of the sampled function $u(x)$ as input and outputs the vector $\boldsymbol{b} = \{b_1, b_2, \cdots, b_m\}$. Subsequently, integrate the trunk and branch networks through the inner product of $\boldsymbol{c}$ and $\boldsymbol{b}$, thereby estimating the state value at point $y$. This can be represented as $\mathcal{G}_\theta(u)(y) = \boldsymbol{c} \cdot \boldsymbol{b}$. Here, $y$ represents a position within the domain of $\mathcal{G}^\dagger(u)$, which could be a temporal variable or spatial location, among others.

Furthermore, in light of specific application scenarios, a series of improved neural operator methods have been successively introduced. For instance, Jin et al. proposed a multi-input neural operator based on tensor products (Jin et al., 2022), and Wang et al. introduced a physics-informed neural op-

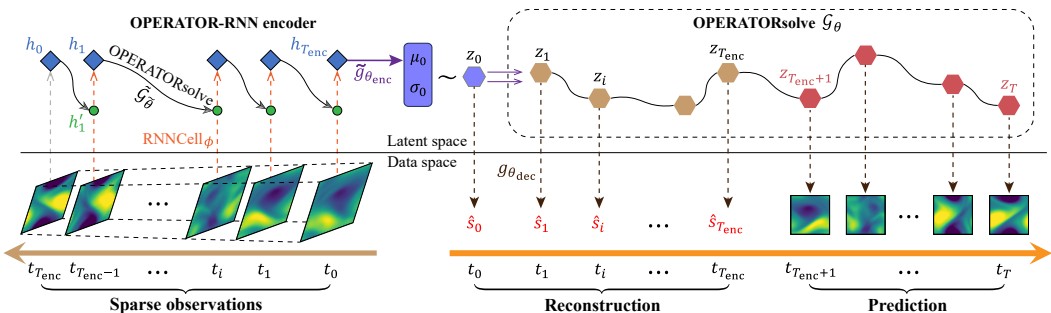

Figure 1: The sketched framework of RLNOs.

erator guided by known dynamical equations (Wang et al., 2021). Additionally, Cao et al. developed a Laplace neural operator capable of handling complex geometrical boundaries (Cao et al., 2024). And Kontolati et al. preliminarily validated the superiority of neural operators in the latent space (Kontolati et al., 2024). Moreover, several studies have endeavored to employ novel neural network architectures for constructing neural operators, achieving superior performance in designated tasks. This includes the use of convolutional neural networks (Raonic et al., 2023), graph neural networks (Sun et al., 2023), and Transformer structures (Hao et al., 2023), among others. These neural operator methods can effectively model a family of dynamical systems, playing a pivotal role across diverse fields such as fluid mechanics (Ye et al., 2024), material science (Oommen et al., 2024), and climate science (Jiang et al., 2023).

## 3 METHOD

In a standard setup, the sampling function $u$ under probability measure $\mu$ and the solution function $s$ forms $N_{\text{tr}}$ training samples $\{u^{(i)}, s^{(i)}\}_{i=1}^{N_{\text{tr}}}$. Then we train parameter $\theta$ using the following equation to obtain the optimal neural operator $\mathcal{G}_{\theta\dagger}$:

$$\min_{\theta \in \Theta} \mathbb{E}_{u \sim \mu} \|\mathcal{G}^\dagger(u) - \mathcal{G}_\theta(u)\| = \min_{\theta \in \Theta} \frac{1}{N_{\text{tr}}} \sum_{i=1}^{N_{\text{tr}}} \|s^{(i)} - \mathcal{G}_\theta(u^{(i)})\| \tag{1}$$

where $\Theta$ is the parameter space of the constructed model, and $\theta^\dagger$ is the optimal parameters obtained post-training. To enhance the robustness of neural operators, we propose the Robust Latent Neural Operator (RLNO) method based on the VAE framework in the latent space. Subsequently, we elaborate on our RLNO method in conjunction with Fig.1.

### 3.1 NEURAL OPERATORS IN LATENT SPACE

Representations in latent spaces are ubiquitous in machine learning tasks, where the system dynamics govern the behavior of latent variables $z$ in latent space, and the system states $s$ are interpreted as manifestations of $z$ under a specific observation function $g$. Given that $g$ is unknown, it can be represented by a neural network $g_{\theta_{\text{dec}}}$ parameterized by $\theta_{\text{dec}}$. Accordingly, we need to establish a neural operator that maps the sampling function $u$ to the solution function $z$ in the latent space, and this operator is still denoted as $\mathcal{G}_\theta$ in our work.

Then we consider the framework of VAE and employ a generative model defined by neural operators to estimate the solution function trajectory:

$$\begin{aligned} z_0 &\sim p(z_0), \\ z_0, z_1, \cdots, z_T &= \text{OPERATORsolve}(\mathcal{G}_\theta, u, z_0, (t_0, t_1, \cdots, t_T)), \\ s_i &\sim p(s_i | g_{\theta_{\text{dec}}}(z_i)), \quad i = 0, 1, \cdots, T, \end{aligned} \tag{2}$$

where $z_0$ signifies the initial state of the latent variable at time $t_0$, sampled from the probability distribution $p(z_0)$. And it is commonly assumed that the prior distribution of $z_0$ adheres to a Gaussian distribution in the VAE framework. Furthermore, "OPERATORsolve" denotes the utilization

of the neural operator $\mathcal{G}_\theta$ to determine the evolution of the latent variable. The decoder $g_{\theta_{\text{dec}}}$ maps these latent variables to the parameters of the probability distribution $p(s_i|g_{\theta_{\text{dec}}}(z_i))$. In DeepONet framework, we initially consider a common scenario where the sample function $u$ corresponds to the initial value $z_0$, then we have

$$z_i = \text{OPERATORsolve}(\mathcal{G}_\theta, z_0, (t_i)) = \mathcal{G}_\theta(z_0)(t_i) = \sum_{k=1}^{m} b_k(z_0)c_k(t_i), \quad i = 0, 1, \cdots, T, \quad (3)$$

where $z$ represents the latent variable in ODE systems. Similarly, for PDE systems, we have

$$z_i = \text{OPERATORsolve}(\mathcal{G}_\theta, z_0, \boldsymbol{x}, (t_i)) = \sum_{k=1}^{m} b_k(z_0)c_k(\boldsymbol{x}, t_i), \quad i = 0, 1, \cdots, T, \quad (4)$$

where $\boldsymbol{x}$ denotes spatial dimension, which takes the form $(x)$ for one-dimensional (1D) PDEs and $(x, y)$ for two-dimensional (2D) PDEs.

## 3.2 ENCODING FOR NON-UNIFORM INTERVAL OBSERVATION DATA

Inspired by the framework of Latent NODEs (Chen et al., 2018), we employ a backward RNN to encode sparse observations into the initial distribution of latent variables, that is, employing the posterior distribution $q(z_0|\{s_i, t_i\}_{i=0}^{T_{\text{enc}}})$ to approximate the distribution $p(z_0)$. However, in practical applications, the observational data $\{s_i, t_i\}_{i=0}^{T_{\text{enc}}}$ may not be uniformly spaced, and the traditional RNN encoder is incapable of encoding the temporal intervals. In earlier work by Rubanova et al., two alternative approaches were introduced (Rubanova et al., 2019). One employs hidden states in RNN that decay exponentially over time, referred to as RNN-Decay encoder, and the other is based on NODE, known as the ODE-RNN encoder. However, the effectiveness of the RNN-Decay encoder requires further enhancement, while the ODE process of the ODE-RNN encoder exhibits significant computational complexity.

---

**Algorithm 1:** OPERATOR-RNN encoder

---

**Require:** Data and corresponding timestamps $\{s_i, t_i\}_{i=0}^{T_{\text{enc}}}$
**Ensure:** Distribution parameters for the initial hidden state $z_0$, i.e., mean $\mu_{z_0}$ and standard deviation $\sigma_{z_0}$
  1: Initialize $h_0 = 0$;
  2: **for** $i$ in $1, 2, \cdots, T_{\text{enc}}$ **do**
      $h'_i = \text{OPERATORsolve}(\tilde{\mathcal{G}}_{\tilde{\theta}}, h_{i-1}, (t_{T_{\text{enc}}-i+1} - t_{T_{\text{enc}}-i}))$;
      $h_i = \text{RNNCell}_\phi(h'_i, s_{T_{\text{enc}}-i})$;
    **end for**
  3: $\mu_{z_0}, \sigma_{z_0} = \tilde{g}_{\theta_{\text{enc}}}(h_{T_{\text{enc}}})$;
  4: **Return**: $\mu_{z_0}, \sigma_{z_0}$

---

Therefore, we design a novel encoder based on neural operators, named OPERATOR-RNN encoder:

$$\begin{aligned} h'_i &= \text{OPERATORsolve}(\tilde{\mathcal{G}}_{\tilde{\theta}}, h_{i-1}, (t_{T_{\text{enc}}-i+1} - t_{T_{\text{enc}}-i})) = \tilde{\mathcal{G}}_{\tilde{\theta}}(h_{i-1})(t_{T_{\text{enc}}-i+1} - t_{T_{\text{enc}}-i}), \\ h_i &= \text{RNNCell}_\phi(h'_i, s_{T_{\text{enc}}-i}), \qquad \text{and } i \in \{1, 2, \cdots, T_{\text{enc}}\}, \end{aligned} \quad (5)$$

where $\tilde{\mathcal{G}}_{\tilde{\theta}}$ denotes a neural operator parameterized by $\tilde{\theta}$, and it is employed in the first equation of the OPERATOR-RNN encoder, thereby incorporating the temporal intervals into the encoder process. Finally, we map the final hidden state $h_{T_{\text{enc}}}$ to the mean $\mu_{z_0}$ and standard deviation $\sigma_{z_0}$ of the distribution $q(z_0|\{s_i, t_i\}_{i=0}^{T_{\text{enc}}})$ through a three-layer neural network $\tilde{g}_{\theta_{\text{enc}}}$ parameterized by $\theta$, i.e., $\mu_{z_0}, \sigma_{z_0} = \tilde{g}_{\theta_{\text{enc}}}(h_{T_{\text{dec}}})$. The pseudocode for the execution process of the OPERATOR-RNN encoder is provided in Algorithm 1, succinctly described as

$$\begin{aligned} q(z_0|\{s_i, t_i\}_{i=0}^{T_{\text{enc}}}) &= \mathcal{N}(\mu_{z_0}, \sigma_{z_0}), \\ \mu_{z_0}, \sigma_{z_0} &= \text{OPERATOR-RNN}_{\theta_{\text{enc}}, \tilde{\theta}, \phi}(\{s_i, t_i\}_{i=0}^{T_{\text{enc}}}), \end{aligned} \quad (6)$$

where $\theta_{\text{enc}}$, $\tilde{\theta}$, and $\phi$ represent the trainable parameters of the OPERATOR-RNN encoder. It should be noted that in the above process, we feed the observational data in reverse order from $t_{T_{\text{enc}}}$ to $t_0$.

### 3.3 TRAINING RLNO USING EVIDENCE LOWER BOUND

Finally, we train the encoder, decoder, and the neural operator concurrently by maximizing the Evidence Lower Bound (ELBO):

$$
\begin{aligned}
\text{ELBO}(\theta_{\text{enc}}, \tilde{\theta}, \phi, \theta, \theta_{\text{dec}}) = \mathbb{E}_{z_0 \sim q_{\theta_{\text{enc}}, \tilde{\theta}, \phi}\left(z_0 | \{s_i, t_i\}_{i=0}^{T_{\text{enc}}}\right)} \left[\log p_{\theta, \theta_{\text{dec}}}(s_0, s_1, \cdots, s_T | z_0)\right] \\
- \text{KL}\left(q_{\theta_{\text{enc}}, \tilde{\theta}, \phi}\left(z_0 | \{s_i, t_i\}_{i=0}^{T_{\text{enc}}}\right) \| p(z_0)\right),
\end{aligned}
\tag{7}
$$

where the first term represents the data likelihood, the second term denotes the Kullback-Leibler (KL) divergence between the prior distribution $p$ and the estimated distribution $q$ of the initial state $z_0$, and the detailed derivation process is provided in Appendix A. It should be noted that the first term comprises $T$ data points, while the OPERATOR-RNN encoder considers only the initial $T_{\text{enc}}$ data points. Given the reality of sparse observations and the need for extrapolation prediction, we typically adopt $T \gg T_{\text{enc}}$. Consequently, the trained RLNO model can effectively perform system reconstruction and prediction tasks.

### 3.4 THE CASE OF MULTI-INPUT FUNCTIONS

Beyond the initial hidden state $z_0$, our approach may incorporate additional sampling functions as inputs, such as parameter functions and boundary conditions. Without loss of generality, we consider an additional sampling function $u$. Following the approach outlined in (Jin et al., 2022), we can extend the RLNO method. Specifically, we introduce a new branch network and incorporate the additional sampling function $u$ as input, and it outputs the vector $\{d_1, d_2, \cdots, d_m\}$. Consequently, the neural operator $\mathcal{G}_\theta$ depicted in Equation (3) can be extended to:

$$
\begin{aligned}
z_i = \text{OPERATORsolve}(\mathcal{G}_\theta, z_0, u, (t_i)) = \mathcal{G}_\theta(z_0)(u)(t_i) \\
= \sum_{k=1}^{m} b_k(z_0) d_k(u) c_k(t_i), \quad i = 0, 1, \cdots, T.
\end{aligned}
\tag{8}
$$

To facilitate the discussion in experiments, we refer to the aforementioned extension method as Multi Input RLNO (MI-RLNO).

## 4 EXPERIMENTS

In this section, we conduct experiments on a system equipped with 64GB RAM and an NVIDIA Tesla V100 GPU with 16GB of memory, and provide a detailed analysis of our methodology across several representative systems. In real-world applications, observational errors are commonly encountered. Therefore, we introduce the Gaussian observational noise with a mean of 0 and a standard deviation of $\sigma_n$ into the simulated experiment data. Additionally, to further validate the effectiveness of our method, we benchmark our experimental results against state-of-the-art (SOTA) baseline methods, namely: Deep Operator Network (DeepONet) (Lu et al., 2021), Multi-Input DeepONet (MI-DON) (Jin et al., 2022), GRU Variational Autoencoder (GRUVAE) (Rubanova et al., 2019), GRU Decay (GRUDecay) (Che et al., 2018), Latent DeepONet with a multi-layer autoencoder (MLAE) (Kontolati et al., 2024), and Latent Neural ODE (LNODE) (Rubanova et al., 2019). Moreover, for systems governed by PDEs, we also consider PDE-Net (Long et al., 2018) and Fourier Neural Operator (FNO) (Li et al., 2020) to provide a comprehensive comparative analysis. And the implementation details and discussion about the above baseline methods can be found in Appendix B. Additionally, we detail the hyperparameter settings of RLNO method in Appendix C.

### 4.1 TOY DATASET

First, we consider a toy dataset of periodic trajectories from the literature on the LNODE approach (Rubanova et al., 2019). Specifically, we take $\sigma_n = 0.2$ and $T = 100$ to generat 2000 trajectories with non-equidistant observations, allocating 80% for training and 20% for testing. The frequency and initial value of each trajectory are sampled from their respective uniform distributions. Subsequently, we set $T_{\text{enc}} = 10$ for recognition network and employ various methods to learn the dynamics of these trajectories, and the experimental results are shown in Fig. 2. It is evident from Fig. 2 that

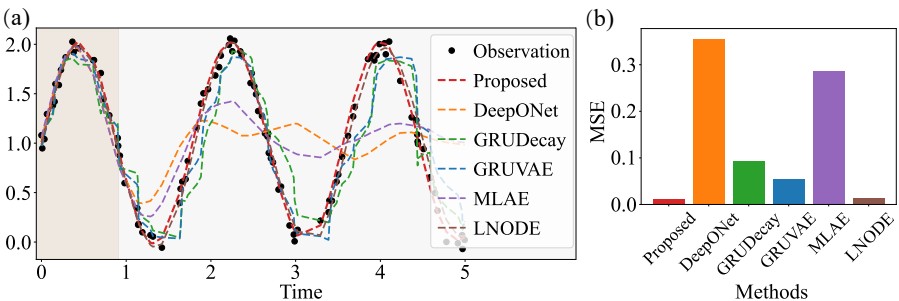

Figure 2: The experimental results of toy dataset using different methods. (a) The prediction performance in a test data. (b) The mean squared error (MSE) of predictions by various methods.

our RLNO method effectively predicts this family of systems, exhibiting the lowest prediction error compared to baseline methods.

Under the conditions of unknown frequencies and high noise, our RLNO method significantly outperforms traditional baseline methods based on RNNs and neural operators, thereby demonstrating the necessity and superiority of our framework design. In addition, although the LNODE method exhibits comparable predictive performance to our method in this toy experiment, it suffers from catastrophic failure in terms of computational cost and training convergence as dimensionality increases, particularly for the PDE systems discussed subsequently.

## 4.2 1D PDE Systems

In this section, we validate our approach through two representative 1D PDE systems, including the Diffusion-Reaction (DR) equation system:

$$\partial_t s = 0.01\partial_{xx}s + 0.01s^2 + u_1(x,t), \ x \in [0,1], \ t \in [0,1], \tag{9}$$

and the Kuramoto-Sivashinsky equation (KS) system:

$$\partial_t s = -\partial_x(s^2/2) - \partial_{xx}s - u_2(x,t)\partial_{xxxx}s, \ x \in [0,2\pi], \ t \in [0,0.5], \tag{10}$$

where $u_1$ and $u_2$ represent the parameter functions of the DR and KS systems, respectively. To more robustly validate our method, we consider constructing a family of dynamical systems and performing experiments through two approaches. The first involves generating initial values $s_0$ using Gaussian random fields (GRFs) with a radial-basis function kernel, given by

$$s_0 \sim c \cdot G \left\{ \mu_G, \exp\left[ ||x_1 - x_2||^2/(2l^2) \right] \right\}, \tag{11}$$

where $\mu_G$ represents the mean value, $l$ is the length scale that governs the smoothness of the sampling function, and $c$ serves as the scaling factor for the output. And the second approach pertains to the random generation of parameters $u$ through GRFs. Additionally, during the data generation phase, each piece of data comprises $T$ equidistant data points. Subsequently, sparse observational data is created by randomly retaining a proportion $\lambda$ of these points. Unless otherwise specified, we set the recognition network parameter $T_{enc}$ to 10.

Table 1: Comparing the MSE ($\pm$ two standard deviations) across multiple experiments

| Systems | (MI-) DON | GRUVAE | GRUDecay | MLAE | LNODE | FNO | (MI-) RLNO |
|---------|-----------|--------|----------|------|-------|-----|------------|
| DR (case 1) | $0.0114_{\pm 0.0146}$ | $0.0255_{\pm 0.0279}$ | $0.0181_{\pm 0.0214}$ | $0.0094_{\pm 0.0119}$ | $0.0033_{\pm 0.0045}$ | $0.0238_{\pm 0.0245}$ | $\mathbf{0.0012}_{\pm \mathbf{0.0016}}$ |
| KS (case 1) | $0.3697_{\pm 0.8492}$ | $3.1764_{\pm 5.1487}$ | $2.8613_{\pm 4.5272}$ | $0.4116_{\pm 0.5953}$ | $0.3212_{\pm 0.7671}$ | $0.4438_{\pm 0.7112}$ | $\mathbf{0.1230}_{\pm \mathbf{0.2325}}$ |
| DR (case 2) | $0.2240_{\pm 0.3976}$ | $0.0080_{\pm 0.0140}$ | $0.0216_{\pm 0.0479}$ | $0.1632_{\pm 0.1888}$ | $0.0219_{\pm 0.0475}$ | $0.2921_{\pm 0.3727}$ | $\mathbf{0.0017}_{\pm \mathbf{0.0020}}$ |
| KS (case 2) | $0.5686_{\pm 0.8873}$ | $5.0330_{\pm 9.1339}$ | $4.9334_{\pm 8.7576}$ | $0.8876_{\pm 1.2794}$ | $0.9740_{\pm 1.3793}$ | $0.7478_{\pm 0.8183}$ | $\mathbf{0.2389}_{\pm \mathbf{0.3637}}$ |
| NS (case 1) | $0.0013_{\pm 0.0009}$ | $0.0030_{\pm 0.0054}$ | $0.0052_{\pm 0.0048}$ | $0.0037_{\pm 0.0055}$ | $0.0015_{\pm 0.0017}$ | $0.0024_{\pm 0.0009}$ | $\mathbf{0.0005}_{\pm \mathbf{0.0004}}$ |
| NS (case 2) | $0.0009_{\pm 0.0017}$ | $0.0291_{\pm 0.0306}$ | $0.0417_{\pm 0.0437}$ | $0.0032_{\pm 0.0041}$ | $0.0026_{\pm 0.0108}$ | $0.0040_{\pm 0.0064}$ | $\mathbf{0.0001}_{\pm \mathbf{0.0002}}$ |

In the first experimental setup, we fix the parameters $u_1 \equiv 0$ and $u_2 \equiv 0.081$, and employ GRFs to generate the initial condition $s_0$, thereby modeling a family of dynamical systems with varying initial

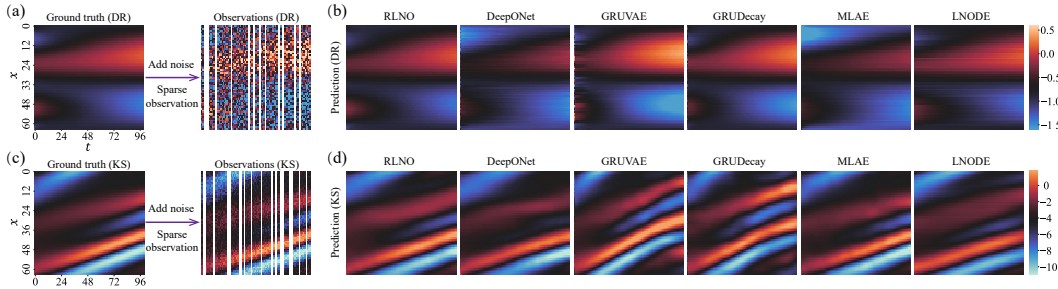

Figure 3: Experimental results of various methods in DR and KS systems. (a) True and observed values of DR system. (b) Prediction results of the DR system. (c) True and observed values of KS system. (d) Prediction results of the DR system.

values. For the experiments of DR system, experimental data were generated based on the following parameter settings: GRF parameters $c = 1.0$, $l = 0.2$, the numbers of training and test samples $N_{tr} = 8000$, $N_{te} = 2000$, the number of sampling points and step size $T = 100$, $\Delta t = 0.01$, sparsity parameter $\lambda = 0.6$, and noise intensity $\sigma_n = 0.4$. Figure 3(a) randomly presents a sample of test data. After training, the predictive performances of RLNO and the baseline methods are depicted in Fig. 3(b), with the corresponding Mean Squared Error (MSE) results reported in the first row of Table 1. For the experiments of KS system, we generate experimental data with the following parameters: $c = 10$, $l = 0.2$, $N_{tr} = 4000$, $N_{te} = 1000$, $T = 100$, $\Delta t = 0.005$, $\lambda = 0.6$, $\sigma_n = 1.0$, and an example of test data is shown in Fig. 3(c). After training, the predictive performance is illustrated in Fig.3(d), with the corresponding MSE results detailed in the second row of Table 1. Given that the KS system exhibits chaotic behavior and is a stiff system, the task of modeling this family of dynamical systems becomes significantly more complex. Despite these challenges, our approach maintains a more robust and accurate predictive performance, significantly surpassing other baseline methods. In this context, even when the LNODE method employs smaller simulation steps and higher-precision numerical solvers, it fails to achieve predictive performance comparable to that of the RLNO method.

To further validate the effectiveness of our approach, two additional experiments are considered. Initially, for the DR system, we fix the initial condition $s_0$ and utilize the GRF to generate the parameter function $u$, thereby modeling a family of dynamical systems with varying parameters. Here, we take $\sigma_n = 0.1$ and maintain all other experimental parameters consistent with those of the DR (case 1). As illustrated in the third row of Table 1, it is evident that our method achieves the lowest prediction error. It is noteworthy that in the experiments of DR (case 2), we configure $u(x)$ as a time-invariant sampling function. In this context, our approach can accurately model the neural operator even without knowledge of the sampling function $u(x)$. This is attributed to the RLNO method's capacity to capture the underlying dynamics related to $u(x)$ via non-uniform sparse observations. In such scenarios, RNN-based methods (GRUVAE and GRUDecay) outperform methods solely based on neural operators (DeepONet, FNO and MLAE). Our proposed RLNO method, which leverages the advantages of both RNN encoding and neural operators, therefore, demonstrates optimal predictive performance.

Finally, we examine a more complex scenario involving the KS system, wherein both the initial value condition $s_0$ and the parameter function $u$ are randomly generated by GRFs. Here, we designate $u(t)$ as a time-varying sampling function, with the corresponding GRF parameters set to $c = 0.01$ and $l = 0.5$, and all other parameters are consistent with those employed in the experiments of KS (case 1). In this scenario, we utilize MI-DON as the first baseline method, and treat both the parameter function $u$ and initial value $s_0$ as inputs to the FNO method. Given that $T_{enc} \ll T$, the OPERATOR-RNN encoder is unable to capture the parameter information across the entire data. Consequently, we employ the MI-RLNO method to incorporate $u(t)$ as an additional input correspondingly. As demonstrated in the fourth row of Table 1, it is evident that our approach yields the most superior predictive performance. Additionally, methods based on neural operators outperform those based on RNNs, attributed to the utilization of $u(t)$ information.

Moreover, we also apply the PDE-NET approach for predicting the PDE systems discussed in this paper. However, we observe that such methods struggle to accurately infer the underlying equations

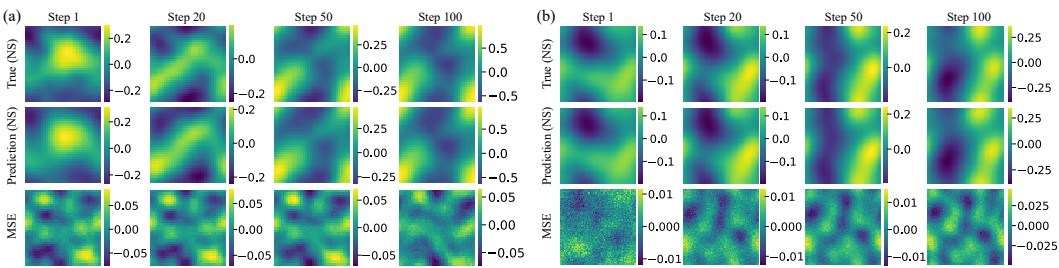

Figure 4: Experimental performance in NS system using RLNO method. (a) Experimental results for NS (case 1). (b) Experimental results for NS (case 2).

of the systems in noisy environments. Consequently, the predictive error tends to escalate rapidly with accumulation, leading to the divergence of predictive outcomes midway. Therefore, this paper does not specifically present the predictive results of the PDE-NET baseline method.

### 4.3    2D NS System

Next, we explore the application of the proposed method within the context of 2D PDE systems, and its dynamic equations are:

$$\partial_t s = \partial_x \gamma \partial_y s - \partial_y \gamma \partial_x s + 0.001 \Delta s + u(x, y, t), \ \ \Delta \gamma = -s, \tag{12}$$

where $(x, y) \in [0, 2]^2$, $t \in [0, 3]$, $\gamma$ is the stream function, $\Delta$ represents the Laplacian operator. Subsequently, we generate experimental data based on the following experimental parameters: $N_{\text{tr}} = 4000$, $N_{\text{te}} = 1000$, $T = 100$, $\Delta t = 0.03$, $\gamma = 0.6$, $\sigma_{\text{n}} = 0.2$.

Similarly, here we conduct experiments under two distinct scenarios. For the first case, we consider a spatial resolution of $32 \times 32$ and employ a 2D GRFs to randomly generate the initial condition $s_0$, thereby modeling a family of systems with varying initial conditions. After training, the experimental results of the RLNO method are illustrated in Fig 4(a), with the predictive MSE of various methods presented in the fifth row of Table 1. For the second case, we consider a spatial resolution of $64 \times 64$, and employ a 2D GRFs to randomly generate initial values $s_0$ and parameters $u$. The corresponding experimental outcomes are presented in Fig. 4(b) and the last row of Table 1. The results from these experiments indicate that our proposed RLNO method is capable of utilizing sparse observations to achieve more robust and accurate modeling.

### 4.4    Ablation Experiments and Robustness Analysis

In this section, we validate the robustness of the RLNO method under varying parameter settings through experiments and demonstrate the advantages of the OPERATOR-RNN encoder and the VAE framework through ablation studies.

Table 2: Comparing the MSE ($\pm$ two standard deviations) across different $\sigma_{\text{n}}$ and $N_{\text{tr}}$

| Methods | $\sigma_{\text{n}} = 0.1$ | $\sigma_{\text{n}} = 0.6$ | $\sigma_{\text{n}} = 1.2$ | $N_{\text{tr}} = 2000$ | $N_{\text{tr}} = 5000$ | $N_{\text{tr}} = 20000$ |
|---|---|---|---|---|---|---|
| Ab1 | $0.0006_{\pm 0.0025}$ | $0.0037_{\pm 0.0068}$ | $0.0115_{\pm 0.0146}$ | $0.0084_{\pm 0.0243}$ | $0.0076_{\pm 0.0107}$ | $0.0072_{\pm 0.0070}$ |
| Ab2 | $\mathbf{0.0005_{\pm 0.0028}}$ | $0.0025_{\pm 0.0066}$ | $0.0097_{\pm 0.0134}$ | $\mathbf{0.0057_{\pm 0.0179}}$ | $0.0053_{\pm 0.0093}$ | $0.0046_{\pm 0.0051}$ |
| Ab3 | $0.0010_{\pm 0.0038}$ | $0.0047_{\pm 0.0103}$ | $0.0147_{\pm 0.0212}$ | $0.0104_{\pm 0.0219}$ | $0.0064_{\pm 0.0138}$ | $0.0052_{\pm 0.0050}$ |
| Ab4 | $0.0008_{\pm 0.0043}$ | $0.0049_{\pm 0.0186}$ | $0.0187_{\pm 0.0324}$ | $0.0198_{\pm 0.0488}$ | $0.0120_{\pm 0.0271}$ | $0.0041_{\pm 0.0046}$ |
| RLNO | $\mathbf{0.0006_{\pm 0.0025}}$ | $\mathbf{0.0023_{\pm 0.0048}}$ | $\mathbf{0.0067_{\pm 0.0114}}$ | $\mathbf{0.0054_{\pm 0.0183}}$ | $\mathbf{0.0041_{\pm 0.0091}}$ | $\mathbf{0.0029_{\pm 0.0038}}$ |

To rigorously validate the effectiveness of our approach, we design the following ablation experiments: First, we modify the RLNO framework by substituting the OPERATOR-RNN encoder with a standard RNN encoder, denoting this variant as "Ab1"; second, within the RLNO framework, we replace the OPERATOR-RNN encoder with the RNN-Decay encoder, and label this adaptation as "Ab2"; third, we replace the OPERATOR-RNN encoder with the ODE-RNN encoder, denoting this

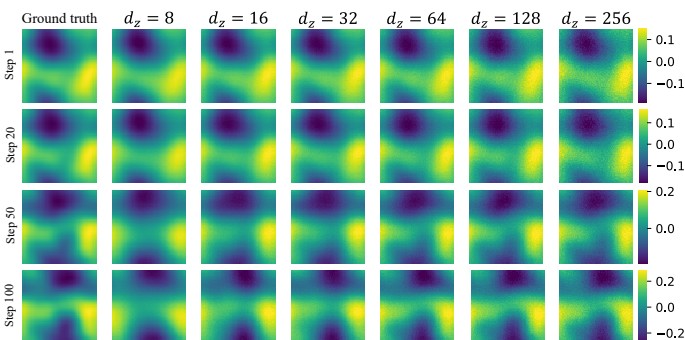

Figure 5: Experimental results across different $d_z$ in a test data of NS (case 2) experiment.

approach as "Ab3"; fourth, we alter the loss function from the ELBO to MSE, which we designate as "Ab4".

First, we conduct experiments of DR (case 1) under varying noise intensities and training set sizes. The predicted MSE is presented in Table 2 and the predicted performance is shown in Figs. S1 and S2 in Appendix C, where the sparsity parameter $\lambda = 0.2$, the first three columns of experiments utilize $N_{\text{tr}} = 4000$, the latter three columns are conducted with $\sigma_{\text{n}} = 1.0$, and all other parameter settings remain consistent with those described previously. Experimental results demonstrate that, in scenarios with low observation noise, various ablation study methods yield commendable outcomes, among which RLNO and Ab2 exhibit optimal performance. As the intensity of noise escalates, the superiority of the RLNO method becomes increasingly pronounced. On one hand, our method exhibits lower prediction errors in comparison with both Ab1 and Ab2, thereby confirming the superiority of the OPERATOR-RNN encoder; on the other hand, Ab3 displays the highest prediction error, corroborating that training using the ELBO indeed possesses enhanced noise robustness. In addition, our approach consistently demonstrates superior performance across varying training set sizes $N_{\text{tr}}$, particularly manifesting significant advantages with abundant training data. It is evident that the OPERATOR-RNN encoder, coupled with the ELBO training objective, can fully leverage training data to learn more authentic underlying dynamics.

Table 3: Comparing the MSE ($\pm$ two standard deviations) across different encoding lengths $T_{\text{enc}}$

| Methods | $T_{\text{enc}} = 1$ | $T_{\text{enc}} = 2$ | $T_{\text{enc}} = 3$ | $T_{\text{enc}} = 6$ | $T_{\text{enc}} = 12$ | $T_{\text{enc}} = 20$ |
|---|---|---|---|---|---|---|
| Ab1 | $0.3868_{\pm 0.6011}$ | $0.3076_{\pm 0.4414}$ | $0.2875_{\pm 0.4163}$ | $0.1675_{\pm 0.3338}$ | $0.1441_{\pm 0.3047}$ | $0.1427_{\pm 0.3327}$ |
| Ab2 | $0.3868_{\pm 0.6011}$ | $0.2714_{\pm 0.4295}$ | $0.2052_{\pm 0.3524}$ | $0.1644_{\pm 0.2949}$ | $0.1438_{\pm 0.2879}$ | $0.1312_{\pm 0.2712}$ |
| Ab3 | $0.3868_{\pm 0.6011}$ | $0.2854_{\pm 0.3218}$ | $0.2368_{\pm 0.4255}$ | $0.1825_{\pm 0.4707}$ | $0.1997_{\pm 0.4132}$ | $0.1912_{\pm 0.6320}$ |
| Ab4 | $0.4098_{\pm 0.9593}$ | $0.3622_{\pm 0.5661}$ | $0.2695_{\pm 0.5768}$ | $0.2546_{\pm 0.3707}$ | $0.2154_{\pm 0.3624}$ | $0.2551_{\pm 0.5151}$ |
| RLNO | $0.3868_{\pm 0.6011}$ | $\mathbf{0.2164_{\pm 0.3872}}$ | $\mathbf{0.1945_{\pm 0.3287}}$ | $\mathbf{0.1329_{\pm 0.3023}}$ | $\mathbf{0.1271_{\pm 0.2456}}$ | $\mathbf{0.1219_{\pm 0.2165}}$ |

Table 4: Comparing the MSE ($\pm$ two standard deviations) across different latent space dimensions

| Systems | $d_z = 8$ | $d_z = 16$ | $d_z = 32$ | $d_z = 64$ | $d_z = 128$ | $d_z = 256$ |
|---|---|---|---|---|---|---|
| DR (case 1) | $0.0025_{\pm 0.0089}$ | $0.0022_{\pm 0.0072}$ | $0.0021_{\pm 0.0063}$ | $0.0021_{\pm 0.0049}$ | $0.0021_{\pm 0.0073}$ | $0.0020_{\pm 0.0067}$ |
| KS (case 1) | $0.4991_{\pm 0.8232}$ | $0.1473_{\pm 0.3068}$ | $0.1390_{\pm 0.2893}$ | $0.1334_{\pm 0.2742}$ | $0.1304_{\pm 0.2518}$ | $0.1368_{\pm 0.2625}$ |
| NS (case 1) | $0.0024_{\pm 0.0027}$ | $0.0019_{\pm 0.0020}$ | $0.0013_{\pm 0.0007}$ | $0.0012_{\pm 0.0007}$ | $0.0013_{\pm 0.0007}$ | $0.0011_{\pm 0.0006}$ |
| NS (case 2) | $4.6\text{e-}4_{\pm 3.6\text{e-}4}$ | $2.8\text{e-}4_{\pm 2.0\text{e-}4}$ | $2.1\text{e-}4_{\pm 1.6\text{e-}4}$ | $1.5\text{e-}4_{\pm 1.3\text{e-}4}$ | $1.4\text{e-}4_{\pm 1.1\text{e-}4}$ | $1.6\text{e-}4_{\pm 1.6\text{e-}4}$ |

Additionally, we conduct experiments of KS (case 1) under the condition of varying recognition lengths ($T_{\text{enc}}$), and keep all other experimental parameters consistent with previous settings. The predicted MSE is presented in Table 3 and the predicted performance is shown in Figure S3 in Appendix C. The experimental results demonstrate that an increase in $T_{\text{enc}}$ enhances the amount of dynamic information encoded in the initial values of the latent variables, thereby improving the modeling effectiveness. Moreover, as $T_{\text{enc}}$ increases, the magnitude of improvement in modeling performance correspondingly decreases. This phenomenon can be explained from two perspectives.

Firstly, the fixed number of RNN neurons entails an upper limit on its capacity for encoding information. Secondly, the intrinsic memory decay attribute of RNNs leads to a gradual forgetting of inputs from the more distant past. Despite this, the utilization of sparse observation information can indeed significantly enhance the modeling performance of neural operators.

Finally, we opt for varying hidden space dimensions $d_z$ and conduct four experiments: DR (case 1), KS (case 1), NS (case 1), and NS (case 2), and the corresponding predictive results are presented in Table 4. The experimental outcomes indicate that despite the relatively high dimensionality of the experimental data spaces (64 dimensions for the DR and KS experiments, $32 \times 32$ dimensions for NS (case 1), and $64 \times 64$ dimensions for NS (case 2)), it is feasible to select lower-dimensional latent spaces while still maintaining high modeling accuracy. Our experiments partially reveal the redundant nature of features within the original data space, suggesting that this dimensionality reduction can decrease the complexity of the task at hand. However, the dimension of the latent space should not be excessively reduced, as the optimal size is contingent upon the complexity of the task itself. For instance, in the more complex experiment of NS (Case 2), modeling effectiveness noticeably improves as $d_z$ increases, up to a point where $d_z \leq 64$, as shown in the Fig. 5 and Table 4. Therefore, choosing $d_z = 64$ for modeling these systems is efficacious, even though this represents a significant reduction from the original data space dimensions ($64 \times 64$). These findings indicate that our approach is capable of extracting critical dynamical information within a family of systems, thereby facilitating its application to even higher-dimensional and more complex PDE systems.

## 5 CONCLUDING REMARKS

In this work, we introduce a novel neural operator method named RLNO, grounded in the framework of VAEs. This method initially encodes the original system states into a latent space. Subsequently, the future states of these latent variables are predicted by modeling a neural operator $\mathcal{G}_\theta$. Finally, a decoder maps the variables from the latent space back to the data space, thus enabling the modeling of the original system. In this process, our model is trained by maximizing the evidence lower bound, and the Gaussian prior assumption within the model enables our method to more effectively handle observations with noise. We conduct experiments across several parametric ODE and PDE systems, and the results demonstrate that our RLNO method surpasses state-of-the-art baseline methods in terms of noise robustness and modeling accuracy.

To enhance the encoding of sparse observational data, our RLNO method incorporates an OPERATOR-RNN encoder. This encoder not only inputs multiple observational data points but also encodes temporal interval information of sequential data, thereby exhibiting superior modeling performance in irregularly spaced observations compared to traditional RNN encoders. Moreover, while both the ODE-RNN encoder and the RNN-Decay encoder are capable of encoding temporal interval information, their performance is inferior to our method, and the ODE-RNN encoder incurs a high computational cost in PDE systems. This assertion was validated through meticulously designed ablation studies, indicating that our method can more effectively utilize sparse and spaced observational data for operator learning. Additionally, we conducted a parameter sensitivity analysis on the dimensionality of the latent space. Results indicate that lower-dimensional latent states achieve impressive modeling accuracy. This dimensionality reduction in the latent space decreases the task's complexity, facilitating the neural operator's ability to capture essential features with limited training data. Consequently, this approach can be extended to modeling tasks of higher-dimensional complex systems. These findings are consistent with the conclusions drawn in (Kontolati et al., 2024).

Certainly, the RLNO method exhibits certain limitations that warrant further investigation. For instance, when the length of the recognition network $T_{\text{enc}}$ is sufficiently large, the predictive performance of our method does not significantly improve. This suggests that the RNN-based encoders may gradually forget information from more distant observations. Thus, developing an encoder with long-term memory capabilities could further enhance the modeling capacity of RLNO. Moreover, current neural operator methods typically excel in modeling a family of dynamical systems under relatively simple sampling distribution scenarios (e.g., Gaussian random fields). Therefore, future efforts are required to devise more effective neural operators capable of modeling systems within more complex sampling distributions. This is of significant practical importance for replacing inefficient numerical simulations with efficient neural operators.

## REPRODUCIBILITY STATEMENT

To foster reproducibility in research, we take several measures to ensure that our findings are transparent and verifiable. First, the manuscript provides detailed descriptions of the RLNO framework utilized throughout the study. Subsequently, we generate experimental data through the code provided in the "code/data" folder within the supplementary material, and present the related parameter settings in the main text and Appendix C. Finally, the detailed implementation procedure for the RLNO method, baseline approaches, and all ablation study codes are provided in the "code/lib" folder, and all executable code files pertaining to the experiments are located in the "code" folder within the supplementary material.

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

# Appendix

## A  THE EXECUTION DETAILS OF THE RLNO METHOD

### A.1  APPROXIMATE POSTERIOR AND EVIDENCE LOWER BOUND

In this section, we provide a detailed description of the approximate posterior distribution $q$ mentioned in the main text, and derive the optimization objective, the Evidence Lower Bound (ELBO).

Given the data $s_{0:T} = \{s_0, s_1, \cdots, s_T\}$ from the original system, our primary interest in variational inference (VI) tasks lies in the posterior distribution of the latent variables, $p(z_0|s_{0:T})$. Evaluating this distribution directly is often challenging or impractical due to the involved marginalization operations, which require integration or summation over all possible configurations of the latent variables—a task that becomes exceedingly complex in high-dimensional spaces. VI addresses this issue by introducing a more tractable distribution, $q(z_0)$, as an approximation of the posterior distribution. Due to

$$
\begin{aligned}
\log p(s_{0:T}) &= \log \int p(s_{0:T}, z_0)\mathrm{d}z_0 = \log \int p(s_{0:T}, z_0)\frac{q(z_0)}{q(z_0)}\mathrm{d}z_0 \\
&= \log \mathbb{E}_{q(z_0)}\left[\frac{p(s_{0:T}, z_0)}{q(z_0)}\right] \geq \mathbb{E}_{q(z_0)}\left[\log \frac{p(s_{0:T}, z_0)}{q(z_0)}\right] = \text{ELBO},
\end{aligned}
\tag{S1}
$$

the ELBO serves as a lower bound for the log evidence, also known as the log marginal likelihood. Through derivation, we can obtain:

$$
\begin{aligned}
\text{ELBO} &= \int q(z_0) \log \frac{p(s_{0:T}, z_0)}{q(z_0)}\mathrm{d}z_0 = \int q(z_0) \log \frac{p(s_{0:T}|z_0)p(z_0)}{q(z_0)}\mathrm{d}z_0 \\
&= \int q(z_0) \log p(s_{0:T}|z_0)\mathrm{d}z_0 - \int q(z_0) \log \frac{q(z_0)}{p(z_0)}\mathrm{d}z_0 \\
&= \mathcal{L}_1 - \mathcal{L}_2.
\end{aligned}
\tag{S2}
$$

In accordance with Equation (6) in the main text, it is evident that the posterior distribution $q(z_0)$ is determined by the observational data $s_{0:T_{\mathrm{enc}}}$ within the recognition network. This relationship can succinctly be expressed as follows:

$$
z_0 \sim q_{\theta_{\mathrm{enc}}, \tilde{\theta}, \phi}(z_0|\{s_i, t_i\}_{i=0}^{T_{\mathrm{enc}}}),
\tag{S3}
$$

where $\theta_{\mathrm{enc}}$, $\tilde{\theta}$, and $\phi$ represent the trainable parameters in the recognition network as defined in the main text. Therefore, we have:

$$
\mathcal{L}_1 = \mathbb{E}_{z_0 \sim q_{\theta_{\mathrm{enc}}, \tilde{\theta}, \phi}(z_0|\{s_i, t_i\}_{i=0}^{T_{\mathrm{enc}}})}[\log p_{\theta, \theta_{\mathrm{dec}}}(s_0, s_1, \cdots, s_T|z_0)],
\tag{S4}
$$

where $\theta$ and $\theta_{\mathrm{dec}}$ respectively represent the trainable parameters of the neural operator $\mathcal{G}_\theta$ in the latent space and the decoder $g_{\theta_{\mathrm{dec}}}$. And $\mathcal{L}_2$ refers to the Kullback-Leibler (KL) divergence:

$$
\mathcal{L}_2 = \text{KL}\left(q_{\theta_{\mathrm{enc}}, \tilde{\theta}, \phi}\left(z_0|\{s_i, t_i\}_{i=0}^{T_{\mathrm{enc}}}\right)\|p(z_0)\right).
\tag{S5}
$$

Through the aforementioned process, we transform the optimization problem of maximizing $\log p(s_{0:T})$ into maximizing its evidence lower bound.

### A.2  THE EXECUTION PSEUDOCODE OF THE RLNO METHOD

To more effectively illustrate the implementation details of the proposed RLNO method, this section presents its pseudocode in Algorithm 2 and Algorithm 3. In testing process, this methodology takes as input the observed data from a new domain, $\{s_i, t_i\}_{i=0}^{T_{\mathrm{enc}}}$, along with a sampling function $u$ from the domain, and outputs direct predictions of the system states $s_\tau$, $\tau \in [t_0, t_T]$. It is crucial to note that the length of the sparse observations $T_{\mathrm{enc}}$ is typically set much shorter than the total data length $T$, i.e., $T_{\mathrm{enc}} \ll T$.

In addition, in our experiments, the term OPERATORsolve denotes the execution of operator computations utilizing the DeepONet framework, RNNCell employs the GRU module, and the symbols $\tilde{g}$ and $g$ denote the usage of a three-layer feedforward neural network, respectively.

---

**Algorithm 2:** The Training Process of the RLNO Method

---

**Require:** Given $N_{\text{tr}}$ training data instances, each instance comprises $T$ observed states $\{s_i, t_i\}_{i=0}^{T}$, along with the corresponding sampling function $u$

**Ensure:** The trained parameters $\theta, \tilde{\theta}, \phi, \theta_{\text{enc}}, \theta_{\text{dec}}$

1: Initialize $h_0 = 0$;
2: **for** $i$ in $1, 2, \cdots, T_{\text{enc}}$ **do**
$\quad h_i' = \text{OPERATORsolve}(\tilde{\mathcal{G}}_{\tilde{\theta}}, h_{i-1}, (t_{T_{\text{enc}}-i+1} - t_{T_{\text{enc}}-i}))$;
$\quad h_i = \text{RNNCell}_\phi(h_i', s_{T_{\text{enc}}-i})$;
**end for**
3: $\mu_{z_0}, \sigma_{z_0} = \tilde{g}_{\theta_{\text{enc}}}(h_{T_{\text{enc}}})$;
4: Sampling $z_0$ from a Gaussian distribution with mean $\mu_{z_0}$ and variance $\sigma_{z_0}^2$;
5: $z_0, z_1, \ldots, z_T = \text{OPERATORsolve}(\mathcal{G}_\theta, u, z_0, (t_0, t_1, \ldots, t_T))$;
6: $\hat{s}_0, \hat{s}_1, \ldots, \hat{s}_T = g_{\theta_{\text{dec}}}(z_0, z_1, \ldots, z_T)$;
7: Use Equation (7) in the main text to calculate ELBO, and use it as the loss function to train parameters $\theta, \tilde{\theta}, \phi, \theta_{\text{enc}}, \theta_{\text{dec}}$;
8: **Return**: After sufficient training, output the trained parameters $\theta, \tilde{\theta}, \phi, \theta_{\text{enc}}, \theta_{\text{dec}}$

---

**Algorithm 3:** The Testing Process of the RLNO Method

---

**Require:** Sparse observations in a new domain $\{s_i, t_i\}_{i=0}^{T_{\text{enc}}}$, sampling function $u$

**Ensure:** Prediction of the system state, $s(\tau)$, for $\tau \in [t_0, t_T]$, where $T_{\text{enc}} \ll T$

1: Initialize $h_0 = 0$;
2: **for** $i$ in $1, 2, \cdots, T_{\text{enc}}$ **do**
$\quad h_i' = \text{OPERATORsolve}(\tilde{\mathcal{G}}_{\tilde{\theta}}, h_{i-1}, (t_{T_{\text{enc}}-i+1} - t_{T_{\text{enc}}-i}))$;
$\quad h_i = \text{RNNCell}_\phi(h_i', s_{T_{\text{enc}}-i})$;
**end for**
3: $\mu_{z_0}, \sigma_{z_0} = \tilde{g}_{\theta_{\text{enc}}}(h_{T_{\text{enc}}})$;
4: Sampling $z_0$ from a Gaussian distribution with mean $\mu_{z_0}$ and variance $\sigma_{z_0}^2$;
5: $z_\tau = \text{OPERATORsolve}(\mathcal{G}_\theta, u, z_0, (\tau)), \tau \in [t_0, t_T]$;
6: $s_\tau = g_{\theta_{\text{dec}}}(z_\tau), \tau \in [t_0, t_T]$;
7: **Return**: $s_\tau, \tau \in [t_0, t_T]$

---

## A.3 Universal Approximation Theorem for RLNO

To theoretically validate the effectiveness of the proposed method, we analogously present the Universal Approximation Theorem for RLNO, as seen in Theorem 1. Subsequently, we briefly provide the proof of this theorem by integrating the proof process of the classic DeepONet method.

**Theorem 1.** *(**Universal Approximation Theorem for RLNO**). Suppose that $\mathcal{X}$ is a Banach Space, $K_1 \in \mathcal{X}$, $K_2 \in \mathbb{R}^d$ are two compact sets in $\mathcal{X}$ and $\mathbb{R}^d$, respectively, $V$ is a compact set in $C(K_1)$. Assume that $\mathcal{G}$ is a nonlinear continuous operator, which maps $V$ into $C(K_2)$. Then for any $\epsilon > 0$, there are positive integers $l$, $m$, continuous vector functions $\boldsymbol{f} : \mathbb{R}^l \to \mathbb{R}^m$, $\tilde{\boldsymbol{f}} : \mathbb{R}^d \to \mathbb{R}^m$, and $x_1, x_2, \ldots, x_m \in K_1$, such that*

$$\left| \mathcal{G}(u)(y) - g_{dec}\left( \langle \boldsymbol{f}\left(u(x_1), u(x_2), \cdots, u(x_l), z_0\right), \tilde{\boldsymbol{f}}(y) \rangle \right) \right| < \epsilon \tag{S6}$$

*holds for all $u \in V$ and $y \in K_2$, where $\langle \cdot, \cdot \rangle$ denotes the dot product in $\mathbb{R}^l$, and*

$$\begin{aligned} z_0 &\sim q(z_0 | \{s_i, t_i\}_{i=0}^{T_{enc}}) = \mathcal{N}(\mu_{z_0}, \sigma_{z_0}), \\ \mu_{z_0}, \sigma_{z_0} &= \textit{OPERATOR-RNN}(\{s_i, t_i\}_{i=0}^{T_{enc}}), \end{aligned} \tag{S7}$$

*where OPERATOR-RNN serves as an encoder for sparse observations $\{s_i, t_i\}$ from a specific new domain in $V$.*

*Proof.* In fact, when the conditions

$$\begin{aligned} g_{\text{dec}}(z) &= z, \\ \boldsymbol{f}\left(u(x_1), u(x_2), \cdots, u(x_l), z_0\right) &= \boldsymbol{f}\left(u(x_1), u(x_2), \cdots, u(x_l)\right) \end{aligned} \tag{S8}$$

are met, Theorem 1 simplifies to Theorem 2 as presented in Reference (Lu et al., 2021). Therefore, by fixing Equation (S8) and based on the proof in Reference (Lu et al., 2021), it is established that there exist neural networks $\boldsymbol{f}$ and $\tilde{\boldsymbol{f}}$ such that Theorem 1 holds. $\qquad\square$

In practical applications, both the OPERATOR-RNN and $g_{\text{dec}}$ are typically trainable neural networks serving as the encoder and decoder, respectively. Generally, the degenerate case represented by Equation (S8) is not considered. Additional encoder and decoder can significantly enhance the robustness and accuracy of neural operators, as can be substantiated by the following three considerations.

- First, the VAE framework transforms the operator modeling issue in the original data space into an operator modeling problem in the latent space, offering enhanced flexibility and generality. For example, traditional DeepONet approaches, which model the dynamics of observed variables directly, may struggle to capture the implicit relationships within high-dimensional data. In contrast, our method, by compressing or expanding observed data into a latent representation, then modeling the dynamical system within this latent space, is more adept at capturing complex temporal sequence characteristics.

- Second, our approach employs variational inference to model latent variables, enhancing its suitability for handling complex data characterized by uncertainty or noise. In fact, ablation studies and comparative experiments with classic baseline methods convincingly support this conclusion.

- Third, an RNN-based encoder (OPERATOR-RNN) can efficiently leverage additional observed states and potential dynamics information in new domains. In contrast, classical neural operator approaches fail to achieve this. In fact, this can be attributed to the fact that the initial value vectors of latent variables can generate system states under specific sampling functions more robustly and accurately through neural operators in the latent space, thereby modeling a family of dynamical systems.

## B Introduction and Discussion of Baseline Methods

In this section, we offer a succinct overview of the deployment of baseline methodologies within our experiments.

First, we consider the original version of the Deep Operator Network method (Lu et al., 2021), herein referred to as DeepONet. This technique is structured around a branch network and a trunk network. The branch network processes $l$ equidistantly distributed sampling points from the parameter function $u(\boldsymbol{x}, t)$, generating a $m$-dimensional vector $\boldsymbol{b} = \{b_1, \ldots, b_m\}$. Conversely, the trunk network accepts the temporal and spatial variables, $t$ and $\boldsymbol{x}$, respectively, yielding another $m$-dimensional vector $\boldsymbol{c} = \{c_1, \ldots, c_m\}$. Consequently, the prediction of the system's state for any given set of inputs is articulated as follows:

$$\mathscr{G}(u)(\boldsymbol{x}, t) = \sum_{k=1}^{m} b_k(u)c_k(\boldsymbol{x}, t) + q,$$

where $q \in \mathbb{R}$ represents a bias term. After training, we can accurately forecast the state value $s(\boldsymbol{x}, t)$ for any sampled function $u$. Additionally, when both the initial value $s_0$ and parameter $u$ vary, we employ the MI-DON approach (Jin et al., 2022). Specifically, we consider introducing a new branch network that takes the initial value $s_0$ as its input and outputs an $m$-dimensional vector $\boldsymbol{d} = \{d_1, \cdots, d_m\}$, thereby facilitating the prediction of the system's future states:

$$\mathscr{G}(s_0)(u)(\boldsymbol{x}, t) = \sum_{k=1}^{m} b_k(u)c_k(\boldsymbol{x}, t)d_k(s_0) + q,$$

where $q$ is a bias term.

Second, we consider GRU Variational Autoencoder (GRUVAE) method (Rubanova et al., 2019), which is fundamentally an RNN prediction method in latent space. Specifically, the approach initially employs an encoder to map $s_0$ to the initial value $z_0$ in the latent space. Subsequently, it predicts the future states of latent variables through the GRU framework, given by:

$$h_i, z_i = \text{GRUCell}_\phi(h_{i-1}, z_{i-1}), \quad \text{and } i \in \{1, 2, \cdots, T\}. \tag{S9}$$

where $\text{GURCell}_\phi$ is GRU module parameterized by $\phi$. And ultimately, decodes the latent variables back to their original states.

Third, we examine the GRU method with an exponential decay in time (Che et al., 2018), henceforth referred to as GRUDecay. The method described shares similarities with the second approach in that it is also based on RNN. However, the distinctive aspect of GRUDecay lies in its consideration of temporal interval information. Specifically, the GRU state update equation in the hidden space is formulated as follows:

$$\begin{aligned} h_i, z_i &= \text{GRUCell}_\phi(h_{i-1}, t_i - t_{i-1}, z_{i-1}) \\ &= \text{GRUCell}_\phi(h_{i-1} \cdot \exp\{-\tau \Delta_t(i)\}, z_{i-1}), \end{aligned} \tag{S10}$$

where $i \in \{1, 2, \cdots, T\}$, $\Delta_t(i) = t_i - t_{i-1}$ represents the time interval.

Fourth, we consider Latent DeepONet with a multi-layer autoencoder (MLAE) (Kontolati et al., 2024). This method is a two-stage training approach within a latent space. In the first stage, the authors pre-train an autoencoder using a multilayer fully connected neural network. This autoencoder is capable of encoding raw spatial data into a latent space. Subsequently, a DeepONet method is trained within this latent space to predict the state of latent variables at any given time. There are two fundamental distinctions between the MLAE method and our RLNO approach. Firstly, our method's OPERATOR-RNN encoder capitalizes on multiple sparse observational data inputs. Secondly, we employ the VAE framework to train all parameters concurrently.

Fifth, we consider Latent Neural Ordinary Differential Equation (Latent NODE) (Rubanova et al., 2019). This approach shares similarities with our RLNO method, with the primary distinctions being two-fold: Firstly, the method utilizes an ODE-RNN encoder for its recognition network; secondly, it employs "ODEsolve" for predicting the states of latent variables. Despite latent NODE method exhibiting strong predictive performance in numerous low-dimensional systems, it encounters unacceptably high computational complexity when applied to large-scale, high-dimensional systems or PDE systems.

Sixth, we consider the PDE-Net approach (Long et al., 2018), which leverages finite differences to approximate spatial derivative terms and uses simple backward Euler for training and testing. In particular, for 2-d PDE systems, this method employs specific convolution kernels to compute

Table S1: Experimental hyperparameters in different systems

| Experiment | $N_{tr}$ | $N_{te}$ | $N_x$ | $T$ | $T_{enc}$ | $\Delta t$ | $l$ | $c$ | $\lambda$ | $\sigma_n$ | $d_z$ | $d_{rec}$ |
|---|---|---|---|---|---|---|---|---|---|---|---|---|
| Toy model | 1600 | 400 | / | 100 | 10 | / | / | / | / | 0.2 | 32 | 32 |
| DR (case 1) | 8000 | 2000 | 64 | 100 | 10 | 0.01 | 0.2 | 1.0 | 0.6 | 0.4 | 64 | 64 |
| DR (case 2) | 8000 | 2000 | 64 | 100 | 10 | 0.01 | 0.2 | 1.0 | 0.6 | 0.1 | 64 | 64 |
| KS (case 1) | 4000 | 1000 | 64 | 100 | 10 | 0.005 | 0.2 | 10 | 0.6 | 1.0 | 64 | 64 |
| KS (case 2) | 4000 | 1000 | 64 | 100 | 10 | 0.005 | 0.5 | 0.01 | 0.6 | 1.0 | 64 | 64 |
| NS (case 1) | 4000 | 1000 | $32 \times 32$ | 100 | 10 | 0.03 | / | / | 0.6 | 0.2 | 64 | 100 |
| NS (case 2) | 4000 | 1000 | $64 \times 64$ | 100 | 10 | 0.03 | / | / | 0.6 | 0.2 | 64 | 100 |

spatial derivatives. Experiments reveal that the method underperforms when it fails to accurately infer the underlying dynamical equations, inevitably leading to significant prediction errors in roll-out forecasts.

Seventh, we consider the Fourier Neural Operator (FNO) approach (Li et al., 2020), which formulates a neural operator by directly parameterizing the integral kernel in Fourier space. In practice, we take the system state $s(\boldsymbol{x}, t)$ and the parameter function $u(\boldsymbol{x}, t)$ at time $t$ as input and directly output the system state $s(\boldsymbol{x}, t + \delta_t)$ at time $t + \delta_t$. In fact, in our experimental setup, this method is unable to achieve super-resolution prediction along the temporal dimension. Consequently, we opt for predicting the system state at sampling points $\{t_0, t_0 + \Delta t, \cdots, t_0 + T\Delta t\}$ with $\delta_t = \Delta t$.

## C  EXPERIMENTAL PARAMETERS SETTINGS

To facilitate the replication of our experiments, in this section, we provide a detailed account of the hyperparameter settings used in the main text. Here, $N_{tr}$ denotes the number of training sets, $N_{te}$ represents the number of test sets, $N_x$ indicates the spatial discretization dimensionality of PDEs, $T$ stands for the number of sampled data points, $T_{enc}$ signifies the length of the encoder network, $\Delta t$ refers to the sampling time step, $l$ and $c$ respectively correspond to the length scale and scaling factor of GRF, $\lambda$ represents the sparsification ratio, $d_z$ denotes the dimensionality of the latent space for the neural operator, and $d_{rec}$ represents the dimensionality of the latent space for the recognition network.

## D  SUPPLEMENTARY EXPERIMENTAL RESULTS

In this section, we supplement the robustness analysis results mentioned in the main text. For detailed information, please refer to Section 4.4 of the main text and Figures S1-S5.

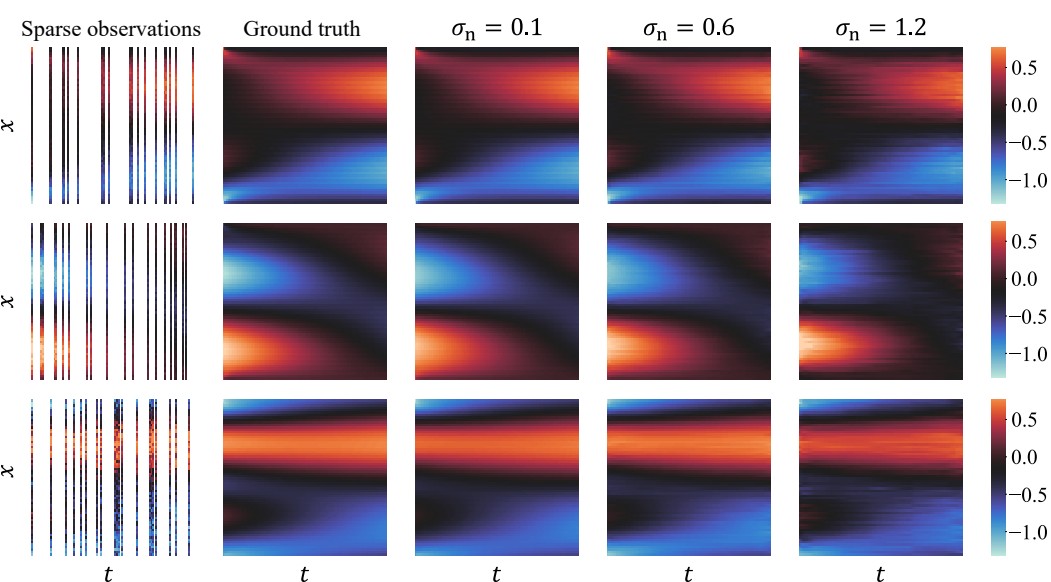

Figure S1: Experimental results across different $\sigma_n$ in three test data of DR (case 1) experiment.

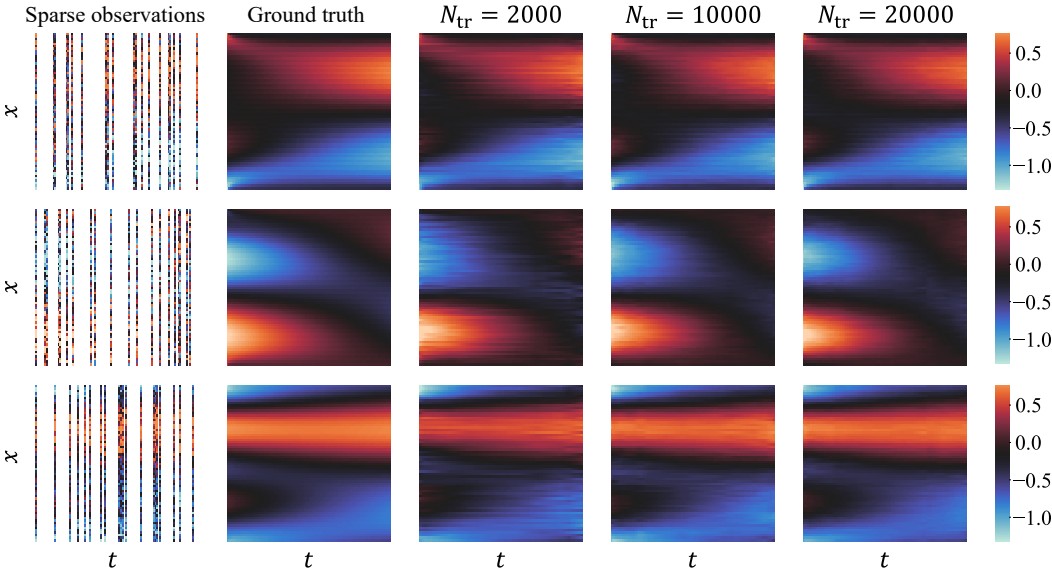

Figure S2: Experimental results across different $N_{tr}$ in three test data of DR (case 1) experiment.

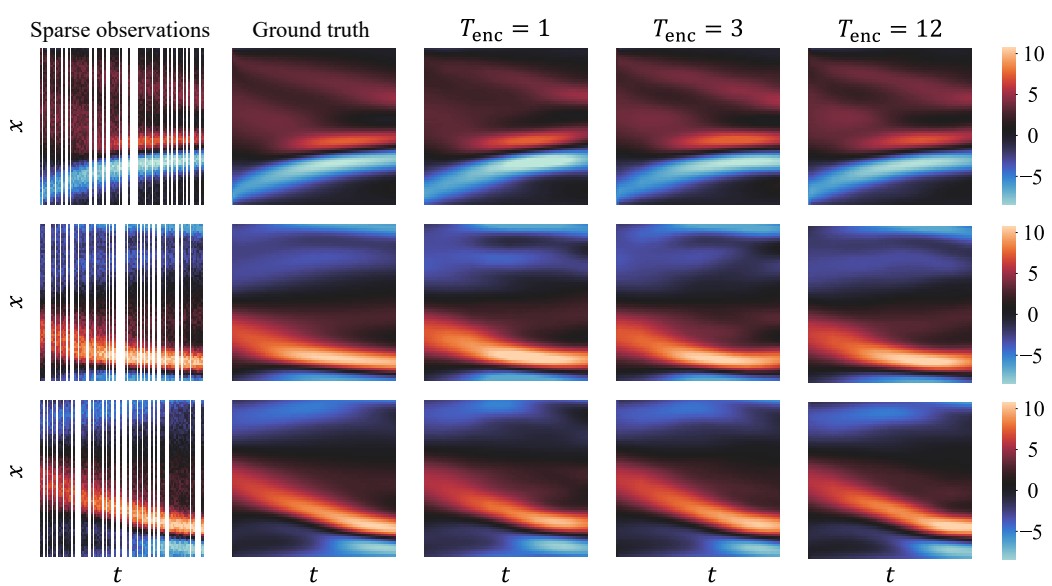

Figure S3: Experimental results across different $T_{\mathrm{enc}}$ in three test data of KS (case 1) experiment.

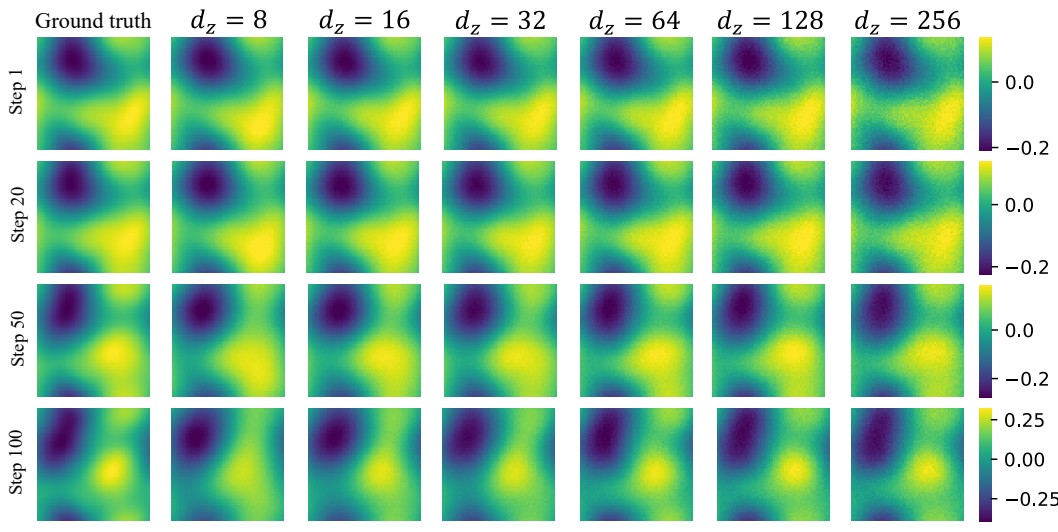

Figure S4: Experimental results across different $d_z$ in a test data of NS (case 2) experiment.

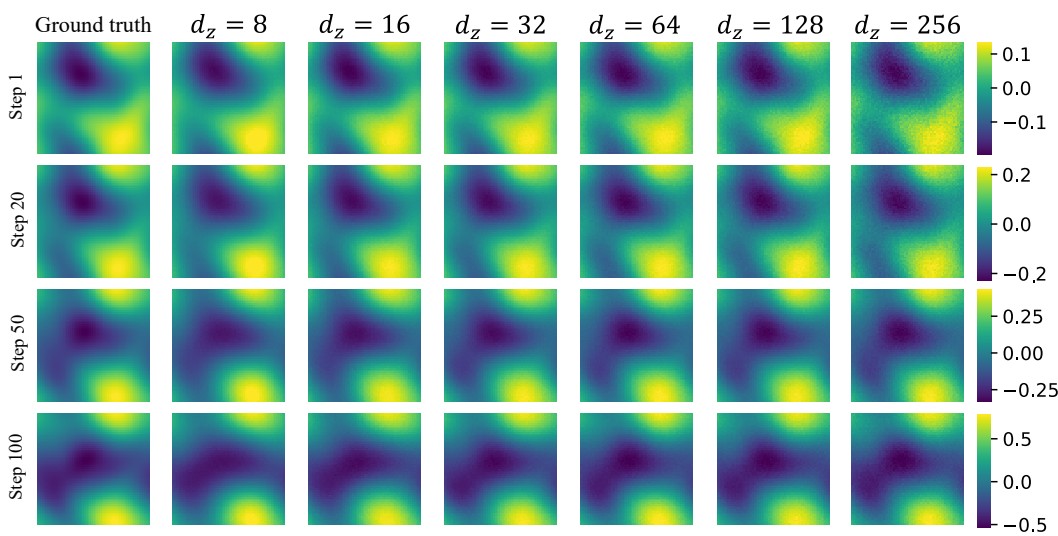

Figure S5: Experimental results across different $d_z$ in a test data of NS (case 2) experiment.

