# OpenReview forum: "Robust Latent Neural Operators for a Family of Systems with Sparse Observations"
_ICLR.cc/2025/Conference — ICLR 2025 Conference Withdrawn Submission_

### Official Review · Reviewer_n6rD · 2024-10-17

**Soundness:** 3
**Presentation:** 2
**Contribution:** 2
**Rating:** 6
**Confidence:** 3

**Summary:**

This paper introduces a latent RNN-based neural operator designed to handle noise and sparse observations in solving partial differential equations (PDEs). The method is trained using a variational auto-encoder framework. While similar frameworks have been explored in literature on irregularly-sampled time series, this work proposes a novel encoder based on DeepONet to address the limitations in effectiveness and efficiency seen in existing approaches. The proposed method is validated on two 1D PDEs (the Diffusion-Reaction equation and the Kuramoto-Sivashinsky equation) and one 2D PDE (the Navier-Stokes equation).

**Strengths:**

This paper addresses a critical limitation of existing neural operators in handling noise and sparse data. The proposed method shows promising results in overcoming these challenges. The experimental setup and data preparation are clearly outlined, and the authors have also provided the implementation code in the supplementary materials.

**Weaknesses:**

The primary contributions of this paper are twofold: 1) the introduction of the latent RNN framework to address noise and sparse data in neural operators, and 2) the proposal of a new encoder based on DeepONet. However, several key issues remain to be addressed:

1. **The rationale behind the latent RNN framework in handling noise and sparse data is insufficiently explained**. I recommend that the authors conduct a more in-depth analysis of the differences between RNN-based methods and existing neural operators to better explain the performance gain of the proposed method. It would be helpful to provide additional insights to further justify the motivation behind the proposed method.

2. **While the DeepONet-based encoder introduces some modifications, its novelty appears limited**. The proposed framework (Equation 2) closely resembles that of [Rubanova et al., 2019] (Equations 5-7), with the primary distinction being the introduction of OPERATORsolve in place of ODESolve. This corresponds to the replacement of existing encoders (RNN-Decay and ODE-RNN) with OPERATOR-RNN. This DeepONet-based encoder is designed to enhance the effectiveness of RNN-Decay in handling irregularly sampled time series and to improve the computational efficiency of ODE-RNN. However, according to the ablation study, the performance improvement of the proposed encoder compared to RNN-Decay is not significant (Ab2 in Tables 2 and 3), especially considering the standard deviation. Additionally, there is no comparison of computational complexity with ODE-RNN to further substantiate the authors’ claims.

3. **The concept of neural operators utilizing latent space has been explored in previous works [Yin et al., 2022; Serrano et al., 2023]**. These studies employed an auto-decoder framework with implicit neural representations to learn the mapping between data space and latent space, as well as to model latent dynamics using Neural ODEs. While their primary focus is on spatially sparse data and temporal extrapolation, their methods appear applicable to the noise and temporally sparse scenarios considered in this paper. The contribution of this work could be further enhanced by comparing the proposed method with these approaches in both spatially sparse and temporally sparse settings.

I am willing to raise the score if the aforementioned issues are effectively addressed.

**Reference**

[Rubanova et al., 2019] "Latent ordinary differential equations for irregularly-sampled time series." NeurIPS 2019.

[Yin et al., 2022] "Continuous pde dynamics forecasting with implicit neural representations." ICLR 2022.

[Serrano et al., 2023] "Operator learning with neural fields: Tackling pdes on general geometries." NeurIPS 2023.

**Questions:**

1. The parenthetical and textual citations are incorrectly used throughout the paper.

2. To provide a more comprehensive ablation study, consider replacing the OPERATOR-RNN encoder with ODE-RNN in an additional experiment. This configuration would be equivalent to Latent ODE [Rubanova et al., 2019]. While the authors have compared their model to Latent ODE in Table 2, the settings used in that comparison differ from those in the ablation study. By including this additional experiment, a direct comparison between OPERATOR-RNN, RNN-decay, and ODE-RNN can be conducted, offering a clearer understanding of their relative performance.

---

> ### Author Response · Authors · 2024-11-16
>
> We thank you for your valuable comments on our work. We kindly recommend that the reviewer reads the "**General response**" for a comprehensive overview of our main novelty and contributions.
>
> ```
> W1: The rationale behind the latent RNN framework in handling noise and sparse data is insufficiently explained.
> ```
> **Response**: Many thanks for your valuable comments. In fact, our approach synergizes the strengths of RNNs and neural operators. Specifically, we employ an OPERATOR-RNN encoder to more efficiently utilize the domain-specific prior information of sparse observations in new test domains, thereby enhancing the robustness and modeling accuracy of neural operators. For more details, please refer to **General response**.
>
> ```
> W2: While the DeepONet-based encoder introduces some modifications, its novelty appears limited.
> ```
> **Response**: Thank you for your valuable comments and helpful suggestions. In fact, the **core contribution** of our work lies in enhancing the robustness and accuracy of neural operators through the extraction of dynamical information from sparse observational data, utilizing an RNN-based approach. It should be noted that the OPERATOR-RNN encoder represents a further development in our work and constitutes a **supplementary contribution**.
>
> In addition, our RLNO method comprises multiple components, each contributing to the prediction accuracy to varying extents. Although employing an RNN-Decay encoder surpasses the performance of the RNN encoder in handling irregularly spaced observational data, there remains room for enhancement. Due to the consistent superiority of the OPERATOR-RNN encoder over the RNN-Decay encoder, it is corroborated that the OPERATOR-RNN encoder can encode sparse observational information into the distribution of the initial latent variables more efficiently.
>
> Finally, RNN methods or Neural ODE approaches require iterative or integral processes during their operation. While a principal design aspect of neural operators circumvents these processes with high computational complexity. Consequently, it is the current consensus among researchers that neural operator methods operate significantly faster than RNN or Neural ODE approaches. Thus, for the sake of conciseness, we opt not to detail this point within the paper.
>
> ```
> The concept of neural operators utilizing latent space has been explored in previous works.
> ```
> **Response**: Many thanks for your insightful comments. In fact, we conduct a comprehensive survey of related research and select RNN-based methods (GRUVAE and GRUDecay), latent neural ODE (LNODE), classical neural operators (DeepONet and FNO), latent neural operators (including MLAE), and PDE-Net as baseline methods for comparative analysis.
>
> Moreover, the essence of our work centers on a neural operator approach. Consequently, it is primarily compared with traditional neural operator methods and previous latent neural operators, and for other types of approaches, we select the **most representative one** as baseline method. The reference [2] you mentioned pertains to a neural PDE approach, which falls under the category of neural ODEs. The reference [3] primarily focuses on operator modeling for complex geometric boundaries, which diverges from the main interest of our work. In other words, our research and the findings of reference [3] can be applied concurrently within a model. Thus, utilizing it as a baseline method does not effectively demonstrate the advantages of our method.
>
> ```
> Q1: The parenthetical and textual citations are incorrectly used throughout the paper.
> ```
> **Response**: Thank you for your careful reading and valuable comments. We meticulously revised both parenthetical and textual citations throughout the manuscript.
>
> ```
> Q2: To provide a more comprehensive ablation study...
> ```
> **Response**: Many thanks for your valuable comments. In fact, we have conducted the ablation experiments you mentioned. In the PDE experiments, we observed that the ODE-RNN encoder operates **exceedingly slowly** and yields mediocre performance. This sluggish performance becomes particularly pronounced in high-dimensional PDE systems. Consequently, we believe the ODE-RNN encoder is an incongruous component within the efficient neural operators. To validate the aforementioned discussion in the main text, we will supplement the results of the ablation experiment on the ODE-RNN encoder in the paper before the 18th.
>
> Finally, we would like to extend our sincere thanks once again for your valuable comments. We hope that our "General response" as well as the individual responses adequately address your concerns and reassess our work. For your convenience, the responses above are provided in a concise and direct manner. Should you have any further inquiries or require additional clarification, we look forward to your response and welcome the opportunity to engage in a more detailed discussion.

---

> > ### Comment · Reviewer_n6rD · 2024-11-21
> >
> > I appreciate the authors' detailed responses, which have addressed most of my concerns. Consequently, I have raised my score to 5. However, several issues remain:
> >
> > ### Major Concerns
> > 1. **Motivation for VAE Framework**
> >    The main contribution of this work lies in integrating the VAE framework with current neural operators. However, the manuscript lacks a detailed introduction to the VAE framework and its relevant literature, which is essential to fully explain the motivation. Specifically, why and how the VAE framework enhances the robustness and accuracy of neural operators should be clearly discussed.
> >
> > 2. **Generalizability with Other Neural Operators**
> >    Since the Robust Latent Neural Operator does not impose explicit constraints on the choice of neural operators, it would be beneficial to demonstrate its application with additional neural operators beyond DeepONet. For example, incorporating methods like the Fourier Neural Operator (FNO) and its improved variants would significantly strengthen the generalizability and contribution of this work.
> >
> > ### Minor Concerns
> > 1. **Implementation Details**
> >    Some critical implementation details are missing. For instance, how are predictions obtained given $s_i$ in Equation 2? Please include a comprehensive description of the training and inference steps, including specific details such as network architectures.
> >
> > 2. **Clarification on Figure 1**
> >    If I understand correctly, OPERATORsolve employs DeepONet to predict $z_i$ given $z_0$ and $t_i$, suggesting it is not auto-regressive (i.e., the prediction of $z_t$ does not depend on $z_{t-1}$). However, Figure 1 depicts the latent embeddings $z_i$ at different steps being derived in an auto-regressive manner, which may cause confusion. This discrepancy should be clarified to avoid misunderstanding.

---

> ### Author Response · Authors · 2024-11-21
>
> Thank you very much for your valuable feedback and for further engaging in the discussion of our work.
>
> ```
> Major Concern 1: Motivation for VAE Framework ...
> ```
> **Response:** Many thanks for your helpful suggestions and valuable comments. In fact, the VAE framework is a crucial component of the proposed RLNO method.
>
> First, the VAE framework transforms the operator modeling issue in the original data space into an operator modeling problem in the latent space, offering enhanced flexibility and generality. For example, traditional DeepONet approaches, which model the dynamics of observed variables directly, may struggle to capture the implicit relationships within high-dimensional data. In contrast, our method, by compressing or expanding observed data into a latent representation, then modeling the dynamical system within this latent space, is more adept at capturing complex temporal sequence characteristics.
>
> Second, our approach employs variational inference to model latent variables, enhancing its suitability for handling complex data characterized by uncertainty or noise. In fact, ablation studies and comparative experiments with classic baseline methods convincingly support this conclusion.
>
> Third, an RNN-based encoder (OPERATOR-RNN) can efficiently leverage additional observed states and potential dynamics information in new domains. In contrast, classical neural operator approaches fail to achieve this. In fact, this can be attributed to the fact that the initial value vectors of latent variables can generate system states under specific sampling functions more robustly and accurately through neural operators in the latent space, thereby modeling a family of dynamical systems.
>
> To enhance clarity regarding the innovative aspects of our paper, we have augmented the above discussion in both the introduction and conclusion sections.
>
> ```
> Major Concern 2: Generalizability with Other Neural Operators ...
> ```
> **Response:** Thank you for your insightful comments. Indeed, our work is not limited to extending the DeepONet approach. Similarly, we can also integrate neural operators such as FNO to present a series of parallel studies. However, to effectively validate the critical role of each component within the proposed RLNO methodology through comparative and ablation studies, our work concentrates on extending the DeepONet approach. We believe that this work has the potential for broader applications and extensions in the future.
>
> ```
> Minor Concern 1: Implementation Details ...
> ```
> **Response:** Many thanks for your careful reading and helpful advice. In Equation 2, we employ a three-layer fully connected neural network as the decoder to map the latent variable $z_i$ back to the original variable $s_i$. To enhance readability, we have added detailed explanations following Equation 2 and supplemented the Appendix with pseudocode illustrating the execution details of our RLNO method.
>
> Additionally, more implementation details regarding the RLNO method, as well as other baseline approaches, can also be found in the code provided in the Supplementary Material.
>
> ```
> Minor Concern 1: Clarification on Figure 1 ...
> ```
> **Response:** Thank you for your valuable comments and helpful suggestions. In fact, our approach represents an advancement of the DeepONet method, thereby inheriting the benefits of neural operator methods. Unlike RNN and neural ODE methodologies, our method does not require iterative steps for system state prediction, thus ensuring exceptionally rapid forecasting capabilities. In other words, we directly compute the states of the latent space by employing  $z_0$, $z_1$, $\cdots$, $z_T$ = OPERATORsolve($\mathcal{G}$, $u$, $z_0$, ($t_0$, $t_1$, $\cdots$, $t_T$)) in the manuscript. The direction of the arrows (from $z_{t-1}$ to $z_t$) in Figure 1 could potentially introduce ambiguity. To prevent any misunderstanding, we have removed the arrows and meticulously refined Figure 1.
>
> Finally, we deeply appreciate the time and effort you have invested in providing valuable feedback and engaging further in discussion. This substantially contributes to the enhancement of our work. We hope that the responses provided above can address your concerns. Should there be any further questions or suggestions, please feel free to raise them for discussion.

---

> > ### Comment · Reviewer_n6rD · 2024-11-25
> >
> > I appreciate the authors’ clarifications on Major Concern 1 and Minor Concerns 1 and 2. After reviewing the updated manuscript, I am willing to raise my score from 5 to 6. To fully address Major Concern 1, I recommend that the authors include a comprehensive introduction to VAEs and their associated literature in the related works section. Furthermore, regarding Major Concern 2, I strongly suggest conducting experiments that integrate additional neural operators beyond DeepONet in the camera-ready version if the paper is accepted.

---

> > > ### Author Response · Authors · 2024-11-25
> > >
> > > Many thanks for your positive feedback and recognition of our work. Your valuable comments have greatly contributed to enhancing the quality of our work.  To better showcase our work, we will further refine the introduction to VAEs and extend the application of our framework to FNOs.

---

### Official Review · Reviewer_CtqP · 2024-10-20

**Soundness:** 2
**Presentation:** 2
**Contribution:** 1
**Rating:** 3
**Confidence:** 3

**Summary:**

In this paper, the authors focus on the challenges of robustness to noise and sparse observation in neural operators. To tackle these issues, they introduce a latent neural operator that builds upon a VAE framework, leveraging recurrent neural networks to extract information from sparse data. The authors employ a neural operator to infer dynamics in the latent space, followed by a decoder to reconstruct the original data. The extensive experiments conducted across various systems demonstrate the superiority of the proposed method compared to existing baselines.

**Strengths:**

1. The focus on robustness to noise and the capacity to handle sparse observations are critical considerations for neural operators, thus it is appreciated that this work focuses on these two important issues.
2. The proposed method, named RLNO, is somewhat new by extending the DeepONet within a VAE framework.

**Weaknesses:**

1. The inference for the posterior distribution of $z_0$ is somewhat unclear. Why do we need to model $ z_0 $ based on so many observations? Should it not be sufficient for $ z_0 $ to depend solely on $ s_0 $? If so, might it be possible to directly mitigate the issue of irregular sampling in the latent space using an operator solver?
2. The preservation of the functional properties of the proposed method needs demonstration. It is uncertain whether the overall framework retains its status as an operator after the incorporation of multiple non-local operators.
3. In Eq. (7), a more detailed description of $p_{\theta,\theta_{dec}}(s_0, s_1, \dots, s_T|z_0)$ would enhance comprehensibility. Additionally, I am confused by the use of $\theta$ and $\tilde{\theta}$, are they same?
4. The manuscript requires improvements in its written presentation. Some writing suggestions are given below.

**Questions:**

Several writing suggestions:
1. Line 129, if Fig.3 should be Fig. 1?
2. Line 236, repeated ``equation’’

---

> ### Author Response · Authors · 2024-11-17
>
> We sincerely appreciate your valuable comments and helpful suggestions. We kindly recommend that the reviewer reads the "**General response**" for a comprehensive overview of our main novelty and contributions. In response to the individual comments, we have thoroughly considered your suggestions and addressed all concerns in a point-by-point manner as outlined below.
>
> ```
> W1: The inference for the posterior distribution of ...
> ```
> **Response**: Many thanks for your valuable comments. In fact, as mentioned in the **General response**, harnessing multiple sparse observations from a new specific domain constitutes the core contribution of our work. For more details, please refer to **General response**.
>
> Moreover, comparative experiments against conventional DeepONet and latent neural operators, along with the ablation studies presented in Table 2, demonstrate that leveraging multiple domain-specific observational data significantly enhances the modeling accuracy and robustness of neural operators.
>
> ```
> W2: The preservation of the functional properties of the proposed method needs demonstration ...
> ```
> **Response**: Thanks for your instructive comment. Similar to the Latent Neural ODE approach, we establishes a novel neural operator in a latent space, fundamentally enhancing the flexibility and generality of dynamic modeling. As such, our approach retains the numerous advantages associated with neural operators. Additionally, our approach encodes sparse observations from a new specific domain into the distribution of initial latent variables $z_0$, thereby leveraging more domain-specific information. As a result, our method exhibits enhanced robustness and accuracy compared to the baseline methods.
>
> ```
> W3: In Eq. (7), a more detailed description of ...
> ```
> **Response**: Thank you for your careful reading and valuable comments. As shown in Fig.1 in the main text, "$\theta$" refers to the parameters of the neural operator $\mathcal{G}$ in the latent space, whereas "$\tilde{\theta}$" denotes the parameters of the OPERATOR $\tilde{\mathcal{G}}$ within the OPERATOR-RNN encoder. Consequently, these two parameters are entirely distinct and are trained concurrently.
>
> To further enhance the readability of our work, we have carefully revised and refined the expressions within the paper.
>
> ```
> W4: The manuscript requires improvements in its written presentation...
> ```
> **Response**: Many thanks for your careful reading and valuable suggestions. We have corrected two typographical errors you mentioned and carefully reviewed and revised the expressions throughout our manuscript. The latest version of the paper will be updated before the 24th.
>
> Finally, we would like to extend our sincere thanks once again for your valuable comments. We hope that our "General response" as well as the individual responses adequately address your concerns and reassess our work. For your convenience, the responses above are provided in a concise and direct manner. Should you have any further inquiries or require additional clarification, we look forward to your response and welcome the opportunity to engage in a more detailed discussion.

---

> > ### Comment · Reviewer_CtqP · 2024-11-24
> >
> > I appreciate the efforts made by the authors during the rebuttal process. Some of my concerns have been addressed. However, I am so sorry that I cannot raise my score, as the current manuscript still falls below the acceptance criteria from my perspective. My W2 regarding "the preservation of functional properties" remains unresolved. This property is crucial for a neural operator, and additional theoretical analysis is required to explore and deepen this aspect. Additionally, the authors mentioned Latent Neural ODE is not a neural operator. Please refer to [1]
> >
> > [1] Neural Operator: Learning Maps Between Function Spaces With Applications to PDEs. JMLR, 2022.

---

> ### Author Response · Authors · 2024-11-25
>
> Thank you very much for your valuable feedback and for further engaging in the discussion of our work. Here, we believe it is necessary to clarify certain facts regarding the neural operator approach, and to further supplement the theoretical analysis of the RLNO method.
>
> **Firstly**, classical neural operators and neural ODEs represent two distinct approaches to modeling dynamic systems, differing significantly in their principles and application scenarios. The neural operator method (including literature [1] you mentioned) is designed to learn mappings between two infinite-dimensional Banach spaces. It enables the direct prediction of system states within new domains for any sampled functions (such as parametric functions, initial conditions, boundary conditions, etc.), without the need for retraining. The Neural ODE approach can be regarded as a continuous extension of residual networks, aimed at modeling the vector fields of dynamical systems. This approach necessitates the use of a differential equations solver to iteratively compute the system states at future instances. In terms of computational efficiency, especially within high-dimensional or stiff systems, Neural ODEs demonstrate significantly slower speeds compared to neural operators. Therefore, the Latent ODE approach, as an extension of Neural ODE in latent space, should not be considered a neural operator.
>
> **Secondly**, Latent ODE represents an extension of the neural ODE approach within latent spaces, achieving significant advantages in modeling dynamical systems. Inspired by this advancement, we have developed the RLNO method, an expansion of neural operators into latent spaces. In fact, the Latent ODE method boasts significant citation and application; however, its original paper [2] does not focus on "the preservation of functional properties".  Neural operators, in pursuit of enhanced solution speeds compared to the neural ODE methodology, inherently sacrifice a degree of interpretability. Moreover, as a deep learning model, neural operators present significant challenges in analyzing function properties, a common issue within the field of deep learning.
>
> **Thirdly**, the proposed RLNO approach, as an extension of the DeepONet method, inherits the favorable properties of neural operator methods. The existing work on latent neural operators [3] and the experiments in our work validate this point from different perspectives. **To further substantiate this point theoretically, we similarly present the Universal Approximation Theorem for RLNO and its proof in Appendix A.3**.
>
> Finally, we would like to extend our sincere thanks once again for the time and efforts you devoted to reviewing our work. We hope these explanations can address your concerns regarding W2. Should you have any further questions, we warmly welcome you to continue raising them and participate in the discussion.
>
> [2] Rubanova Y, Chen R T Q, Duvenaud D K. Latent ordinary differential equations for irregularly-sampled time series[J]. Advances in neural information processing systems, 2019, 32.
>
> [3] Kontolati K, Goswami S, Em Karniadakis G, et al. Learning nonlinear operators in latent spaces for real-time predictions of complex dynamics in physical systems[J]. Nature Communications, 2024, 15(1): 5101.

---

> ### Comment · Reviewer_CtqP · 2024-11-26
>
> Thank you for the further explanation. After rechecking the submission and response, I now fully recognize the contribution of enhancing robustness by encoding sparse observations into a better initialization value (more precisely, an initialization value distribution). However, I still have several issues:
>
> ### 1. **Preservation of Functional Properties for Neural Operators**
> To verify this, the authors need to theoretically analyze the impact of introducing the Encoder (OPERATOR-RNN, as presented in Eqn. 6) on the functional properties of the overall framework. Specifically, RLNO should be able to learn a solution function independent of the resolution of the input function. This aligns with the statement in the last paragraph of Page 2 in reference [3] provided by the authors:  *"However, for more general problems where training data vary in fidelity ... methods can be employed to create a **shared continuous basis** onto which the training data can be projected"*.
>
> ### 2. **Utilization of Sparse Observations**
> This point is related to my W1. I don't know if encoding subsequent sparse observations into the initialization distribution is an efficient way of utilizing these observations. Could the authors provide a comparison with [3], where subsequent observations directly influence the following inference via neural controlled differential equations [NCDE]? If time does not permit conducting detailed experiments, a clear explanation of the advantages and limitations of the proposed approach compared to NCDE would also be appreciated.
>
> [NCDE] Neural Controlled Differential Equations for Irregular Time Series. NeurIPS, 2020.

---

> ### Author Response · Authors · 2024-11-26
>
> We sincerely appreciate the reviewers for dedicating their valuable time and effort to continue this discussion. Below, we offer detailed explanations in response to the two new questions raised by the reviewer.
>
> ```
> 1. Preservation of Functional Properties for Neural Operators
> ```
> **Response:** Many thanks for your valuable comments and helpful suggestions. In fact, our approach inherits the advantageous properties of the DeepONet method and possesses greater robustness and modeling accuracy. To avoid any misunderstanding, we respond to this issue from the following four points.
>
> **First**, RLNO method inherits the universal approximation theorem of DeepONet, with the related theorem and proof seen in Appendix A.3 of the revised manuscript. Furthermore, our work transforms the operator modeling issue from the data space into the latent space, offering enhanced flexibility and generality. For example, traditional DeepONet approaches, which model the dynamics of observed variables directly, may struggle to capture the implicit relationships within high-dimensional data. In contrast, our method, by compressing or expanding observed data into a latent representation, then modeling the dynamical system within this latent space, is more adept at capturing complex temporal sequence characteristics.
>
> **Second**, RLNO method inherits the super-resolution properties of the DeepONet method, thereby fulfilling the requirement that "RLNO should be able to learn a solution function independent of the resolution of the input function". As shown in Equation (4) of the main text, our approach conducts operator computations in the latent space by directly feeding the spatial variable $\mathbf{x}$ and temporal variable $t$ into the trunk network. This enables prediction of the system state at any position within a specific spatiotemporal domain, independent of the input function's resolution. This property is consistent with the traditional DeepONet approach.
>
> **Third,** The cited paragraph in reference [3] epitomizes the **inability** of neural operator methods to directly address more generalized problems, such as scenarios where training data vary in fidelity **(It is important to note that this issue is distinct from the one discussed in the second point)**. As stated in this paragraph in Reference [3], addressing this issue necessitates the integration of additional linear or nonlinear projection techniques for preprocessing, and the authors state that "the implementation of such methods is beyond the scope of this work". The information provided above clearly indicates that current neural operator methods cannot directly address this issue. In fact, addressing this issue is not an inherent attribute of existing neural operator approaches; rather, it requires the integration of other techniques for preprocessing.
>
> **Fourth,** in the field of deep learning, providing theoretical explanations from a mathematical perspective is a universally acknowledged challenge. Our OPERATOR-RNN encoder, integrating neural operators and RNN architecture, constitutes a deep learning framework, thereby posing challenges to extensive theoretical analysis. To validate the effectiveness of our proposed method, we conduct comparative analyses against multiple baseline methods alongside a series of ablation studies, thereby confirming the indispensability of each component within the RLNO framework. In addition, we have also provided some theoretical analyses in the first and second points, hoping to address your concerns.
>
> ```
> 2. Utilization of Sparse Observations
> ```
> **Response:** Thank you for your careful reading and valuable comments. Firstly, we have already compared with the method in reference [3], referred to as MLAE. In fact, we conduct a comprehensive survey of related research and select RNN-based methods (GRUVAE and GRUDecay), latent ODE (LNODE), classical neural operators (DeepONet and FNO), latent neural operators (MLAE), and PDE-Net as baseline methods for comparative analysis.
>
> Moreover, the essence of our work centers on the neural operators. Consequently, it is primarily compared with traditional neural operator methods and previous latent neural operators, and for other types of approaches, we select the **most representative one** as baseline method. The reference you mentioned pertains to a neural CDE approach that integrates the methods of RNN and Neural ODE, which falls under the category of neural ODEs. In fact, experimental results, especially those derived from experiments involving stiff or PDE systems, indicate that employing neural operators significantly enhances execution efficiency compared to the neural ODEs. This advantage has contributed to the increasing attention on neural operator methods in recent years.
>
> Finally, we hope the above responses can address your concerns. Should you have further questions regarding some specific details, please feel free to raise them for further in-depth discussion.

---

> ### Author Response · Authors · 2024-11-26
>
> **Some Supplementary Clarifications**
> ```
> 1. OPERATOR-RNN encoder
> ```
> In our work, the **core contribution** lies in enhancing the robustness and accuracy of neural operators by leveraging a Variational Autoencoder framework combined with an RNN-based encoder to extract dynamic information from sparse observational data in new domains. It should be noted that the OPERATOR-RNN encoder represents a further development in our work and constitutes a **supplementary contribution**.
>
> Furthermore, to further validate the effectiveness of the OPERATOR-RNN encoder, we designed a series of ablation experiments, which included replacing it with an RNN encoder, an RNN-Decay encoder, and an ODE-RNN encoder. Experimental results confirmed the superiority of the OPERATOR-RNN encoder in leveraging sparse observational data and its underlying dynamical information from these sequential observations.
>
> ```
> 2. Comparative Analysis with NCDE
> ```
> The most significant advantage of our approach over the NCDE method primarily lies in its execution speed. This advantage also underscores the superiority of neural operator methods when compared to RNNs, Neural ODEs, and traditional numerical solving techniques.
>
> In addition, experiments have revealed that error accumulation occurs in the recursive prediction process of Neural ODE methods, including NCDE. Consequently, in situations where training data is relatively abundant, the predictive performance of Neural ODEs is inferior to that of our proposed RLNO method. Although selecting higher-precision numerical solvers, such as "dopri5", for Neural ODE methods can enhance prediction accuracy, it significantly increases the computational cost of training and tesing in PDE systems, rendering it impractically expensive.
>
> Finally, we would like to express our sincere gratitude for the valuable time and effort you have dedicated to reviewing our work. We hope that our most recent two responses can adequately address your concerns. Should you have any further questions or require additional clarification, please do not hesitate to raise them for additional discussion.

---

> ### Comment · Reviewer_CtqP · 2024-11-27
>
> I appreciate the efforts made by the authors during the rebuttal process. What I consistently question about is whether the overall framework still maintains the functional property for approximating operators after introducing the Encoder (OPERATOR-RNN).
>
> The authors claimed that it does. However, based on Theorem 1 provided in Appendix A.3, the answer appears to be **no**. The conditions for the establishment of this proposition is: $\mathcal{f}(u(x_1),u(x_1),\dots,u(x_l), z_0 )= \mathcal{f}(u(x_1),u(x_1),\dots,u(x_l))$, which implies that the encoder makes no contribution to the solution function. This clearly contradicts the main claim of the paper, which asserts that encoding certain observations benefits the final solution function.
>
> To address this issue, I think the authors need to provide proof that, after the introduction of the OPERATOR-RNN, the overall framework still maintains the approximation capability for infinite-dimensional operators defined on the state space. Or decouple the encoding and operator learning process as [3], and rephrase the universal approximation for operators defined solely on the latent space.
>
> [3] Kontolati K, Goswami S, Em Karniadakis G, et al. Learning nonlinear operators in latent spaces for real-time predictions of complex dynamics in physical systems[J]. Nature Communications, 2024, 15(1): 5101.

---

> ### Author Response · Authors · 2024-11-27
>
> Thank you very much for providing specific feedback on your concerns and for engaging in further discussion. We can address this issue in the following four aspects.
>
> 1. The Universal Approximation Theorem of RLNO is an **existence theorem**; hence, we adopted a simplified proof approach by assigning specific values to both OPERATOR-RNN and $g_{\text{dec}}$. In additong, we have elucidated in our prior response that the RLNO method inherits the property of super-resolution prediction from the DeepONet approach.
>
> 2. The core of OPERATOR-RNN is an RNN process, with the "OPERATOR" component designed to address the issue of irregular intervals in observational data. **It is imperative to note that RNNs possess a remarkable capacity for processing temporal data. They not only have the ability to retain memory but are also capable of learning sequence dynamics**. Therefore, the primary function of this encoder is to encode sparse observations in new domains into the initial values of a latent space, leveraging not only the state information from multiple sparse observations but also extracting and utilizing the underlying dynamical information inherent within these observation sequences. **For instance**, in the periodic orbit experiment illustrated in Figure 2, there is no need for additional input of frequency information. Instead, this frequency information can be automatically learned from sparse observations in the new domain. This serves as empirical evidence of OPERATOR-RNN's capability to extract and utilize dynamical information.
>
> 3. From an information theory perspective, our approach incorporates a more substantial amount of information. The conventional DeepONet method inputs only the initial value $s_0$ and parameter function $u$, whereas our RLNO method inputs $u$ as well as encodes the sparse observations $\\{s_0,\dots,s_{T_\text{enc}}\\}$ (including $s_0$) through OPERATOR-RNN for the input. Therefore, a well-trained RLNO model can, at a minimum, achieve better results than the traditional DeepONet approach. This is also the line of reasoning we employed to prove Theorem 1. **For example**, the presence of noise in the input $s_0$ can significantly impact the traditional DeepONet method. In contrast, our RLNO method, utilizing a variational autoencoder framework, is capable of substantially mitigating the effects of noise by considering the multiple observations $\\{s_0,\dots,s_{T_\text{enc}}\\}$.
>
> 4. **The two examples provided in points 2 and 3 precisely underscore the primary reasons for the effectiveness of our RLNO approach**. Therefore, the introduction of the OPERATOR-RNN encoder merely enhances the network's expressive capability using more information, particularly in handling complex, noisy time-series data, without altering the operator approximation framework of DeepONet.
>
> Finally, we acknowledge that providing rigorous mathematical proofs for deep neural network models remains a formidable challenge widely recognized in the field of artificial intelligence. This is particularly true in understanding why the encoding of additional information yields a stable advantage. **For example**, currently, numerous studies, including those involving large models, highlight the importance of incorporating physical information into the model. This emphasis is based on the understanding that additional information renders the model more aligned with real-world scenarios or tasks. Intuitively, this approach makes sense. **However, similar to the issue under our discussion, there is an absence of a rigorous mathematical proof demonstrating the superiority of the deep neural network models embedding additional physical information from the perspective of functional properties**.
>
> We would like to extend our sincerest gratitude for the valuable time and effort you have invested in enhancing the quality of our work. Should you have any additional questions or require further clarification on any aspects, we are more than willing to engage in further discussions.

---

> ### Comment · Reviewer_CtqP · 2024-11-27
>
> Once again, I sincerely thank the authors for their efforts during this rebuttal process. I would like to suggest three references that may help strengthen the theoretical aspects of the submission [1,2,3]. Given the theoretical gap in the current submission, I will maintain my original score.
>
> [1] Recurrent Neural Networks Are Universal Approximators. ICANN, 2006.
>
> [2] Universal approximation to nonlinear operators by neural networks with arbitrary activation functions and its application to dynamical systems. IEEE Transactions on Neural Networks, 1995.
>
> [3] Neural Operator: Learning Maps Between Function Spaces With Applications to PDEs. JMLR, 2022.

---

### Official Review · Reviewer_wUXE · 2024-10-30

**Soundness:** 3
**Presentation:** 1
**Contribution:** 2
**Rating:** 6
**Confidence:** 3

**Summary:**

This paper proposes the Robust Latent Neural Operator (RLNO) that utilizes a recurrent neural network (RNN) encoder and the variational autoencoder (VAE) framework within a latent neural operator learning. The authors claim that the proposed model is more robust to the observation noise and learns the dynamics of a system more accurately.

**Strengths:**

This work introduces mainly two extensions of an existing work, Latent Neural ODE (LNODE) solver. First, it extends the LNODE solver to the operator learning framework. Second, the ODE-RNN encoder of LNODE is replaced with OPEARATOR-RNN in which the neural operator is used instead of an ODE solver. The idea is straightforward, and the proposed integration of each component looks natural and reasonable. The experiments show that the improvement of the proposed model is significant.

---

**After the discussion**, the contributions of the work were getting better recognizable; thus, I raised my score to 6 from 5.

**Weaknesses:**

The extensions rely on existing ideas (such as neural operator/DeepONet, latent space neural ODE, VAE/Evidence Lower Bound loss); thus, the manuscript needs more thorough revision for a clearer presentation of its technical novelty. For example, Algorithm 1 is almost identical to the work of [Rubanova et al. 2019]; thus, it would be worth highlighting the distinctive novelty.

The experiments were evaluated mostly for the "interpolation" task. The "extrapolation" evaluation should be further discussed as shown in the related work (e.g., LNODE).

**Questions:**

For OPERATORSolve, why are the two different neural operators (trained for different parameters $\theta$ and $\tilde{\theta}$) used? Does the proposed method train and use them independently for forward and backward evolution? If so, why?

The extrapolation evaluation should be further clarified. As described, the DR and KS experiments used the sparse observation data sampled from 100 steps of ground truth data and added noise to them. The predictions are evaluated and analyzed for 100 steps. What would the performance be if the prediction happens longer for, e.g., 200 steps and 400 steps? Alternatively, what if the sparse observation data is sampled only at the beginning of the ground truth data?

Minor questions should be clarified:
- L129: The text refers to Fig.3. Is this correct? Maybe Fig. 1?
- L194: Shouldn't it be "encoder" instead of "decoder"?
- L259: What is the RCDI method?
- L329: The text refers to Figure 3(c). Is this correct? Maybe Figure 3(b)?

Minor corrections are required:
- L57: typo "adetly"
- L99: missing space "structuresHao"
- L236: duplicated word "Equation equation 3"
- L309: typo "parse"

---

> ### Author Response · Authors · 2024-11-17
>
> We thank you for your valuable comments on our work. We kindly recommend that the reviewer reads the "**General response**" for a comprehensive overview of our main novelty and contributions.
>
> ```
> W1: The extensions rely on existing ideas (such as neural operator ...
> ```
> **Response**: Many thanks for your valuable comments. In fact, our work draws inspiration from the concept of LNODE in several aspects, yet fundamentally, these two tasks are entirely distinct. Our work represents an advancement in neural operators, a method that circumvents the integration process inherent in neural ODEs, thereby enabling exceedingly rapid prediction of high-dimensional or PDE systems. Furthermore, since neural operators facilitate mapping between two Banach spaces, essentially modeling a family of dynamical systems, they can directly predict system states in new domains without the need for retraining the model.
>
> In Algorithm 1, the primary distinctions between our work and that of [Rubanova et al. (2019)] are twofold. First, we employ neural operator prediction in the latent space, as opposed to neural ODE. Second, we utilize an OPERATOR-RNN encoder instead of an ODE-RNN encoder. Experimental results demonstrate that, notably in PDE systems, our method not only showcases enhanced precision over the LNODE approach but also features a significant speed improvement. This underscores the inherent advantages of neural operator methods over neural ODE approaches.
>
> For details on the main novelty and contributions of our work, please refer to the **General response**.
>
> ```
> W2: The experiments were evaluated mostly for the "interpolation" task ...
> ```
> **Response**: Thank you for your insightful comments. In fact, we have a keen interest and familiarity with techniques such as NODEs, RNNs, and neural operators, each of which emphasizes different aspects in the modeling of dynamical systems. In experiments with neural operators, functions (such as parameter functions, initial boundary conditions, etc.) are typically sampled from a specified distribution, followed by predicting the system's state under the sampled function. Upon training with a limited dataset, the trained neural operators can predict the system state under any sampled function from the distribution. The above process is precisely the task that the neural operator method aims to accomplish.
>
> To the best of our knowledge, current neural operator methods perform poorly in predicting system states outside the temporal and spatial domains of PDE systems or for sampling functions beyond the distribution. This is understandable, as the rapid prediction of neural operators for a family of systems inherently result in the loss of extrapolation ability you mentioned. Indeed, this represents a challenge inherent to all neural operator methods. Despite this, neural operator methods still find numerous critical applications, benefiting from their rapid and precise predictive capabilities.
>
> ```
> Q1: For OPERATORSolve, why are the two different neural operators ...
> ```
> **Response**: Thank you for your careful reading and valuable comments. As shown in Fig.1 in the main text, "$\theta$" refers to the parameters of the neural operator $\mathcal{G}$ in the latent space, whereas "$\tilde{\theta}$" denotes the parameters of the OPERATOR $\tilde{\mathcal{G}}$ within the OPERATOR-RNN encoder. Consequently, these two parameters are entirely distinct and are trained concurrently.
>
> ```
> Q1: The extrapolation evaluation should be further clarified ...
> ```
> **Response**: Many thanks for your valuable comments. Indeed, the task of neural operators focuses on the rapid and accurate prediction of a family of dynamical systems, rather than the extrapolation predictive capabilities you mentioned. For more details, please refer to **the Response in W2**.
>
> In the configuration of our study, when dealing with test data from a new domain, it is crucial to note that **the length of the sparse observations $T_{\text{enc}}$ is typically set much shorter than the total data length $T$, i.e., $T_{\text{enc}}\ll T$** (as shown in Fig.1). In other words, **all our experiments involve sparse observations that are sampled at the beginning of the ground truth data**.
>
> ```
> Q2: Minor questions and corrections should be...
> ```
> **Response**: Many thanks for your careful reading and helpful suggestions. We have corrected the typographical errors you mentioned and carefully reviewed and revised the expressions throughout our manuscript.
>
> Finally, we would like to extend our sincere thanks once again for your valuable comments. We hope that our "General response" as well as the individual responses adequately address your concerns and reassess our work. For your convenience, the responses above are provided in a concise and direct manner. Should you have any further inquiries or require additional clarification, we look forward to your response and welcome the opportunity to engage in a more detailed discussion.

---

> > ### Comment · Reviewer_wUXE · 2024-11-25
> >
> > I appreciate the authors making a great effort in revising the paper and answering my questions.
> >
> > My concerns have been mostly addressed. Nonetheless, I would like to listen to further clarification on the claim on extrapolation. If I understood the claim correctly, the model was trained with sparse observations $T_\mathrm{enc}$ (<< $T$) sampled at the beginning of the trajectory and evaluated for prediction for a larger time interval (i.e., $T$). (Please correct me if I misunderstood. In my view, this is a kind of extrapolation evaluation for predicting time series far beyond the training interval.) Then, for the toy example of Fig. 2, it seems that the model was trained with the observation samples in the dark region ($t \in [0, 1))$ and predicted for a longer time interval (up to $t=5$). (Please indicate what the dark region means.) If so, what are the black (observation) dots in $t=[1,5]$ representing? A similar presentation exists in the other figures (e.g., Fig. 3 and Fig. S1-S3), which confuses me.
> >
> > Additionally, this work may be helpful:
> > - Zhu et al., "Reliable extrapolation of deep neural operators informed by physics or sparse observations," Computer Methods in Applied Mechanics and Engineering, 412, 2023, https://doi.org/10.1016/j.cma.2023.116064.

---

> ### Author Response · Authors · 2024-11-25
>
> Many thanks for your valuable feedback and for further engaging in the discussion about our work.
>
> In fact, as mentioned in responses of W2 and Q2, direct extrapolation tasks over time present challenges for all DeepONet and its extension methods, as they do not iteratively predict in the manner of RNNs or neural ODE approaches. In our work, **the training process necessitates requires $T$ data points to calculate the loss function, whereas the testing phase only requires $T_\text{enc}$ data points as inputs to the OPERATOR-RNN encoder, thereby predicting the system state at any given moment within the interval $[t_0,t_T]$ (here, $T_\text{enc}\ll T$)**. This configuration is indeed rational, as our primary focus is on swiftly predicting the system's state in new domains without the necessity of retraining the model. In addition, to more clearly present the training and testing processes of the RLNO method, we have supplemented Appendix A.2 of the revised manuscript with Algorithms 2 and 3, providing the pseudocode of our method.
>
> In addition, we have carefully reviewed the literature you mentioned and found it to be a remarkably interesting work. As anticipated, this paper does not extend its focus to extrapolation tasks beyond the temporal or spatial domains. Instead, it focuses on extrapolation tasks concerning the parameter $l$ in Gaussian random field (GRF) sampling, where different $l$ values are selected for training and testing sets. As delineated in Section 2 of this paper, when the value of $l$ in the test set is reduced, corresponding to a less smooth sampling function, the extrapolation prediction error of DeepONet significantly increases. This further corroborates our previous assertion regarding the challenges DeepONet faces in extrapolation tasks.
>
> Certainly, the paper you mentioned presents two fine-tuning techniques to alleviate issues faced by neural operators in extrapolation tasks. We believe this solution is justifiable as it employs additional prior information to fine-tune the DeepONet framework. However, the objectives of this study diverge from those of our work, hence we did not specifically consider such extrapolation experiments. In fact, these two fine-tuning techniques are implemented by introducing extra additive loss function terms, making such fine-tuning techniques independent and parallelizable with our RLNO method. Therefore, in future work, we can directly integrate these fine-tuning techniques into the RLNO method to further enhance the extrapolative capability of our approach.
>
> Finally, we deeply appreciate the time and effort you have invested in providing valuable feedback and engaging further in discussion. This substantially contributes to the enhancement of our work. We hope that the responses provided above can address your concerns. Should there be any further questions or suggestions, please feel free to raise them for discussion.

---

> > ### Comment · Reviewer_wUXE · 2024-11-26
> >
> > I appreciate the authors' further clarification. Now, I better recognize the contributions of the work. Thus, I raised my score to 6 from 5.
> >
> > If the paper is accepted, I recommend that the authors include a comprehensive discussion on extrapolation.

---

> > > ### Author Response · Authors · 2024-11-26
> > >
> > > Many thanks for raising the score and your recognition of our work. We will integrate the detailed discussion on extrapolation into our revised version.

---

### Official Review · Reviewer_vdLf · 2024-11-04

**Soundness:** 2
**Presentation:** 2
**Contribution:** 2
**Rating:** 5
**Confidence:** 4

**Summary:**

In this work, the authors introduce RLNO, a robust latent neural operator grounded in the VAE framework, to address the drawbacks of neural operator methods that lack robustness and accuracy due to their sensitivity to noise and inability to handle sparse data. RLNO comprises an RNN-based encoder to capture sequential and dynamical information from sparse observations, a latent space neural operator for modeling, and a decoder to reconstruct the original system. Experimental results across ODE and PDE systems verify RLNO's superiority in both accuracy and robustness.

**Strengths:**

The work presents a derivation of an evidence lower bound that transforms the optimization problem from maximizing observation probability to maximizing the bound, which provides a different approach to the optimization process. The study includes experimental evaluations across multiple test cases.

**Weaknesses:**

- The major limitation of this work lies in its insufficient justification for latent space dynamics inference, particularly since DeepONet already provides established methods for iterative dynamics inference. This design choice lacks clear comparative advantages over existing approaches.

- Though the encoder-decoder architecture is introduced to address sample complexity and sparse observations, the paper falls short in demonstrating how these components preserve DeepONet's fundamental functional properties. This gap raises concerns about the theoretical soundness of the proposed framework.

- The current manuscript structure and presentation hinder a clear understanding of the methodology and contributions.

**Questions:**

please see above.

---

> ### Author Response · Authors · 2024-11-17
>
> We thank you for your valuable comments on our work. We kindly recommend that the reviewer reads the "**General response**" for a comprehensive overview of our main novelty and contributions. Additionally, the main improvements of this paper are detailed in **General Response 2**.
>
> ```
> W1: The major limitation of this work lies in its insufficient justification...
> ```
> **Response**: Many thanks for your valuable comments. In fact, modeling neural operators in latent space offers three main advantages. **Firstly**, modeling the dynamics of the latent space to predict the original system states enhances the flexibility and generality of the constructed model. Historically, numerous machine learning approaches have employed this strategy, including our two baseline methods: Latent Neural ODE and Latent Neural Operator. **Secondly**, we can encode domain-specific sparse observations from a new domain into the distribution of the initial latent variables $z_0$, thereby leveraging additional prior information to achieve more robust and precise modeling. **Thirdly**, by selecting an appropriate dimensionality for the latent space, the complexity of the task can be modulated. For instance, in certain tasks, opting for a lower-dimensional latent space can still preserve high predictive accuracy, thereby reducing task complexity and facilitating the neural operator's ability to capture essential features with limited training data. Consequently, this approach can be extended to higher-dimensional PDE systems.
>
> Additionally, from an experimental standpoint, our approach demonstrates significant enhancements in both robustness and prediction accuracy when compared to multiple baseline methods including DeepONet, thereby validating the effectiveness of our methodology.
>
> For more information on the novelty and contributions of our method, please refer to the **General Response** or the revised Introduction.
>
> ```
> W2: Though the encoder-decoder architecture is introduced to address sample complexity...
> ```
> **Response**: Thank you for your careful reading and valuable comments. Here, we need to make the following three clarifications. **Firstly**, our approach not only inherits the advantages of DeepONet in modeling a family of dynamical systems (**including super-resolution and universal approximation theorem**), but also enhances the flexibility and generality of neural operator modeling by transitioning the modeling process to a latent space. For example, traditional DeepONet approaches, which model the dynamics of observed variables directly, may struggle to capture the implicit relationships within high-dimensional data. In contrast, our method, by compressing or expanding observed data into a latent representation, then modeling the dynamical system within this latent space, is more adept at capturing complex temporal sequence characteristics. **Secondly**, our approach does not aim to solve the issue of sparse observations; rather, it capitalizes on the sparse observational data in new domains, along with extracting underlying dynamical information from the sparse observational sequences, thereby enhancing the modeling accuracy and robustness of neural operators. **Thirdly**, Our work builds upon a foundation of numerous highly effective studies, such as DeepONet, Latent neural ODE, and Latent neural operators, each component being essential and meaningful. To corroborate this assertion, we also conducted ablation studies, thereby thoroughly substantiating the indispensability of our OPERATOR-RNN encoder, ELBO optimization objective, and the configuration of the latent space, among other components. Additionally, to ensure the reliability and reproducibility of our experimental outcomes, we provide all code related to our method, baseline methods, and ablation experiments in the supplementary material.
>
> ```
> W3: The current manuscript structure and presentation hinder a clear understanding of the methodology and contributions.
> ```
> **Response**: Thank you for your constructive feedback and valuable comments. For the specific information on the novelty and contributions of our work, please refer to the "**General Response**". In addition, to prevent misunderstandings and further enhance the readability of the manuscript, we have carefully revised and refined the expressions throughout the paper and enriched the details in Introduction section and Fig. 1. And the main improvements of this paper are detailed in **General Response 2**.
>
> Finally, we would like to extend our sincere thanks once again for your valuable comments. We hope that our "General response" as well as the individual responses adequately address your concerns and reassess our work. For your convenience, the responses above are provided in a concise and direct manner. Should you have any further inquiries or require additional clarification, we look forward to your response and welcome the opportunity to engage in a more detailed discussion.

---

### Author Response · Authors · 2024-11-16

**General response:**

We would like to thank the reviewers for your time, efforts, and valuable comments and suggestions, which do help us to significantly improve the quality of this work. Due to potential shortcomings in readability of our manuscript, we have found that the reviewers **may have some misunderstandings about our method**. Hence, we succinctly summarize the **novelty** and **contributions** of our work as follows.

(1) **Compared to traditional neural operator approaches, our method can utilize more domain-specific information.** Current neural operator methods, including DeepONet, often only input sampled functions (such as parameter functions, initial-boundary conditions, etc.) when tested in new domains. However, in many practical applications, there may exist prior observational data at several moments in this new domain, which is often overlooked in previous methods. Our research has found that fully leveraging these sparse prior observational states has a very significant effect on promoting the robustness and accuracy of neural operator modeling. This benefit arises from domain-specific observations, which not only provide more state information but also extract specific dynamic information from these sequential observations.

(2) **Embedding sparse observations from new domains into neural operators requires a sophisticated framework design.** To fully excavate and utilize the dynamic information within sparse observations, we develop a novel learning framework based on a variational autoencoder and adopt an RNN-based encoder. This method fully leverages the advantages of RNNs and neural operators to better model a family of dynamical systems.

(3) **Design an OPERATOR-RNN encoder to better handle unevenly spaced sparse observation data.** Here, we draw inspiration from the design of the ODE-RNN encoder and design an OPERATOR-RNN encoder, which can more robustly and efficiently extract domain-specific dynamic information from sparse observational data. It should be noted that in terms of computational efficiency, our method inherits the advantages of neural operators and has a very significant advantage over methods such as Latent ODE.

(4) **Our method can select a smaller latent space dimension, thereby reducing the complexity of the operator modeling task.** Results indicate that lower-dimensional latent states achieve impressive modeling accuracy. This dimensionality reduction in the latent space decreases the task's complexity, facilitating the neural operator's ability to capture essential features with limited training data. Consequently, this approach can be extended to modeling tasks of higher-dimensional complex systems. In addition, our approach exhibits higher robustness and accuracy compared to previous latent neural operator methods, attributable to point (1).

In summary, every component within our method is essential and valuable. To substantiate this claim, we select a variety of baseline methods (including approaches based on RNNs, classical neural ODEs, classical neural operators and latent neural operators), and conducted numerous ablation studies to affirm the significant contribution of each component in our methodology. Additionally, to enhance the reliability and reproducibility of our experimental results, we also provide all the experiment codes in the supplementary materials. Moreover, to prevent misunderstandings and further enhance the readability of the manuscript, we have carefully revised and refined the expressions throughout the paper and enriched the details in Fig. 1.


**General response 2:**

We have meticulously addressed all the issues and suggestions of the reviewers and have successfully **submitted a revised manuscript**. Specifically, **we have made the following enhancements:**

* 1. We have supplemented and refined the introduction section, succinctly summarizing the innovative aspects of our work.
* 2. We have corrected typographical errors as pointed out by the reviewers and conducted a thorough revision to enhance the clarity and expressiveness of the entire manuscript.
* 3. We have revised Figure 1 and its associated description, thereby increasing the readability of the method in the paper.
* 4. We have refined the details of our methodology in the main text and augmented Appendix A.2 with the pseudocode for the proposed RLNO method to enhance the readability of the paper.
* 5. We have supplemented our paper with ablation experiments on the ODE-RNN encoder to more effectively validate the robustness and accuracy of our approach.
* 6. We have supplemented the Universal Approximation Theorem for RLNO, as well as elucidated the significant role of the Variational Autoencoder framework, detailed in Appendix A.3.

Finally, we thank all the reviewers again for your valuable and insightful comments. We hope that our General response as well as the individual responses for each reviewer adequately addresses the reviewers’ concerns.

---

### Note · Authors · 2025-01-24

I have read and agree with the venue's withdrawal policy on behalf of myself and my co-authors.